# Projective Equivariant Networks via Second-order Fundamental Differential Invariants

**Yikang Li[1], Yeqing Qiu[2,3], Yuxuan Chen[4], Lingshen He[1], Lexiang Hu[1], Zhouchen Lin[1,5]\***
[1]State Key Lab of General AI, School of Intelligence Science and Technology, Peking University
[2]School of Science and Engineering, The Chinese University of Hong Kong, Shenzhen
[3]Shenzhen Research Institute of Big Data
[4]Khoury College of Computer Sciences, Northeastern University
[5]Institute for Artificial Intelligence, Peking University
liyk18@pku.edu.cn, yeqingqiu@link.cuhk.edu.cn, chen.yuxuan7@northeastern.edu
lingshenhe@pku.edu.cn, hulx@stu.pku.edu.cn, zlin@pku.edu.cn

## Abstract

Equivariant networks enhance model efficiency and generalization by embedding symmetry priors into their architectures. However, most existing methods, primarily based on group convolutions and steerable convolutions, face significant limitations when dealing with complex transformation groups, particularly the projective group, which plays a crucial role in vision. In this work, we tackle the challenge by constructing projective equivariant networks based on differential invariants. Using the moving frame method with a carefully selected cross section tailored for multi-dimensional functions, we derive a complete and concise set of second-order fundamental differential invariants of the projective group. We provide a rigorous analysis of the properties and transformation relationships of their underlying components, yielding a further simplified and unified set of fundamental differential invariants, which facilitates both theoretical analysis and practical applications. Building on this foundation, we develop **PDINet**, the first framework for deep projective equivariant networks, achieving full projective equivariance without discretizing or sampling the group. Empirical results on the projectively transformed STL-10 and Imagenette datasets show that PDINet achieves improvements of 11.39% and 5.66% in accuracy over the respective standard baselines under out-of-distribution settings, demonstrating its strong generalization to complex geometric transformations.

## 1 Introduction

Incorporating symmetry as an inductive bias into neural networks has emerged as a powerful approach to enhance model efficiency and generalization. Convolutional neural networks (CNNs) [Krizhevsky et al., 2012, Simonyan and Zisserman, 2015, He et al., 2016, Chen et al., 2017], which are among the most widely used architectures in deep learning, owe much of their success to the inherent translational equivariance. Building on this idea, Cohen and Welling [2016a] proposed Group Equivariant CNNs (G-CNNs), which generalize equivariance to broader transformations like rotations and reflections. Equivariant networks achieve symmetry incorporation by constructing network layers whose outputs transform in a predictable pattern under group actions applied to the inputs.

The development of equivariant networks began with G-CNNs, whose feature map can be seen as a function on a group. Although G-CNNs have proven effective in various tasks [Worrall and Brostow, 2018, Esteves et al., 2019, Lafarge et al., 2021, Shamsolmoali et al., 2021], they are less suited

---

*Corresponding author.

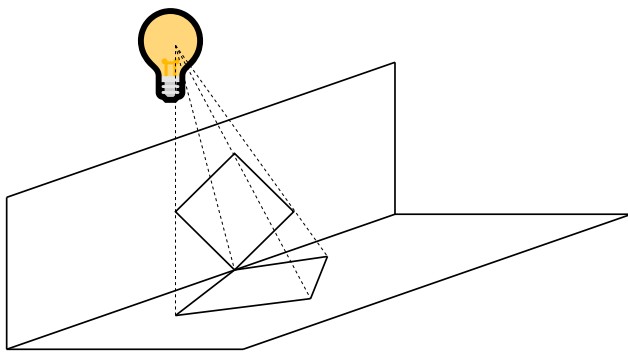

Figure 1: An illustration of a projective transformation: when a planar object is projected onto another plane, a projective transformation occurs.

to continuous groups, as handling such groups typically requires group sampling or discretization, which introduces approximation errors and computational complexity. Then, steerable CNNs [Cohen and Welling, 2016b, Weiler and Cesa, 2019] were proposed to overcome these limitations by viewing features as fields that transform according to specified group representations. In this framework, G-CNNs can be interpreted as a special case where the group representation is chosen to be the regular representation. Steerable CNNs are capable of handling continuous groups such as $SO(2)$ and $SO(3)$ directly, thereby significantly broadening the scope of equivariant networks to include real-world symmetries beyond discrete groups [Weiler et al., 2018, Wang et al., 2020, Wang and Walters, 2022]. However, for more complex non-compact Lie groups such as the projective group, deriving closed-form steerable basis filters becomes intractable, inherently limiting the applicability of steerable CNNs. To address this, MacDonald et al. [2022] enabled group convolutions over finite-dimensional Lie groups by computing the integral on the Lie algebra, thus introducing a projective equivariant model, homConv. However, this method still relies on group sampling, which results in exponential memory growth with increasing network depth, thereby hindering scalability to deeper architectures. Mironenco and Forré [2024] improved sampling efficiency via group decompositions, but focused solely on affine subgroups like $\mathbb{R}^n \rtimes GL^+(n, \mathbb{R})$ and $\mathbb{R}^n \rtimes SL(n, \mathbb{R})$, without addressing more complex groups such as the projective group. Recently, Li et al. [2024, 2025] proposed InvarLayer and steerable EquivarLayer that construct affine equivariant networks based on invariants, enabling closed-form and sampling-free affine equivariance. However, these works remain specialized to the affine group and do not yet generalize to more complex non-compact groups like the projective group.

Actually, projective transformations play a fundamental role in computer vision [Mohr and Triggs, 1996, Birchfield, 1998, Hartley and Zisserman, 2003], as they capture the relationships between objects and their images under perspective projections (see Figure 1). Achieving equivariance on projective transformations is especially critical in practical applications such as mobile robot navigation, 3D scene analysis, and camera pose estimation, where accurately handling perspective effects and viewpoint changes can significantly enhance model robustness and accuracy [Lee et al., 2000, Hartley and Zisserman, 2003, Mur-Artal et al., 2015, Schönberger et al., 2016]. Early on, Suk and Flusser [2004] proposed a projective invariant feature extraction method based on projective moment invariants. Nevertheless, the invariants are formulated as infinite series of moment products, leading to significant computational overhead and intractable error analysis in practical implementations. To overcome this limitation, Li et al. [2018] proposed an alternative framework that constructs projective invariants using finite combinations of weighted moments, where the weights are derived from relative projective differential invariants. Note that moment-based projective invariants are essentially global image descriptors, which makes them inappropriate for constructing equivariant operators that act locally on feature fields for capturing fine-grained spatial patterns. Instead, differential invariants inherently hold the property of acting locally at each spatial position, which makes them a natural foundation for building equivariant operators [Sangalli et al., 2022, 2023, Li et al., 2024, 2025]. While Olver [2023] has proposed a systematic framework for the computation of projective differential invariants via the moving frame method, it requires at least third-order derivatives because

the projective action is not free at second order for scalar functions. Besides, they only consider single-channel cases, and the extension to multi-dimensional cases needs more complex expressions involving high-order derivatives, which limits their practicality in computation and applications on color images.

In this work, we construct projective equivariant networks based on differential invariants, achieving full projective equivariance without relying on group discretization or sampling. This overcomes the depth limitations of homConv [MacDonald et al., 2022] and enables effective scaling to deeper architectures. A core challenge lies in deriving concise and practical projective differential invariants. To support color images and multi-channel intermediate features in modern neural networks, we focus on the differential invariants for multi-dimensional functions. In this case, the projective group acts freely on the second-order jet space, allowing us to derive a complete set of second-order fundamental differential invariants using the moving frame method [Olver, 2015], which can express any second-order invariant of the projective group. However, the choice of cross section used to define the moving frame significantly affects the form of the resulting invariants. While a direct extension of the cross section in [Olver, 2023] to the multi-dimensional case is theoretically valid, it leads to prohibitively long expressions with hundreds of terms, rendering them impractical. Instead, we propose a new cross section tailored to the multi-dimensional structure, which involves up to second-order derivatives, yielding a much more concise set of fundamental differential invariants. Further analysis reveals that these fundamental invariants are composed of a set of simpler components. By exploring the algebraic properties and transformation relationships of these components, we further simplify invariants into a unified set of fundamental invariants, facilitating both practical use and theoretical analysis. Based upon these simplified invariants, we design learnable equivariant operators by combining them with parameterized multi-layer perceptrons (MLPs), and embed the operators into standard neural network backbones to build **PDINet**, the first framework for deep projective equivariant networks free from group sampling. Empirical evaluations under challenging out-of-distribution settings demonstrate the strong generalization ability of our model to complex geometric transformations.

We summarize our main contributions as follows:

- We employ the moving frame method to derive a complete set of second-order fundamental differential invariants of the projective group for multi-dimensional functions, enabling support for color images and multi-channel features.

- We conduct an in-depth analysis of the algebraic structure and transformation properties of these invariants, resulting in a further simplified and unified set of fundamental invariants that facilitate both theoretical understanding and practical computation.

- We develop **PDINet** based on second-order projective differential invariants. It is the first time that deep networks achieve full projective equivariance without relying on group discretization or sampling, thus allowing effective scaling to deeper architectures.

- Numerical experiments on projectively deformed STL-10 and Imagenette[2] under out-of-distribution settings demonstrate the effectiveness of our model, with improvements of 11.39% and 5.66% over the standard baseline results, showcasing its strong generalization capability under complex geometric transformations.[3]

## 2 Method

### 2.1 Basic concepts and notations

To begin with, we introduce some basic concepts and notations necessary for our formulation. An image can be viewed as a continuous function $\mathbf{u}(x, y)$ defined on a 2D plane. For example, an RGB image corresponds to a three-dimensional function. Likewise, intermediate features in neural networks can also be interpreted as functions, and each layer can be seen as an operator that maps one function to another.

A central concept in this work is equivariance. If the output of an operator undergoes a corresponding transformation when the input is transformed, it is referred to as equivariance. The formal definition is as follows:

---

[2]Imagenette is a publicly available dataset downloaded from https://github.com/fastai/imagenette.

[3]Our code is available at https://github.com/Liyk127/PDINet.

**Definition 1** *An operator $\psi : \mathcal{F}_1 \to \mathcal{F}_2$ is said to be **equivariant** with respect to a group $G$ if*

$$g \cdot \psi(\mathbf{u}) = \psi(g \cdot \mathbf{u}), \quad \forall g \in G, \mathbf{u} \in \mathcal{F}_1, \tag{1}$$

*where $\mathcal{F}_1$ and $\mathcal{F}_2$ are the input and output function spaces, respectively.*

Let $X$ denote the domain, $U = \mathbb{R}^n$ be the range of a function, and $U^{(d)} = U \times U_1 \times \cdots \times U_d$ be the derivative space up to order $d$. A group action $g \cdot \mathbf{x}$ on the domain naturally induces an action on functions, defined as $(g \cdot \mathbf{u})(\mathbf{x}) = \mathbf{u}(g^{-1} \cdot \mathbf{x})$, which models how geometric transformations deform images. This action further extends to derivatives through prolongation to the jet space $X \times U^{(d)}$. For example, the first-order prolongation of the action on $X \times U^{(1)}$ can be expressed as: $(\mathbf{x}, \mathbf{u}(\mathbf{x}), \nabla\mathbf{u}(\mathbf{x})) \mapsto (\tilde{\mathbf{x}}, \tilde{\mathbf{u}}(\tilde{\mathbf{x}}), \nabla\tilde{\mathbf{u}}(\tilde{\mathbf{x}}))$, where $\tilde{\mathbf{x}} \triangleq g \cdot \mathbf{x}$ and $\tilde{\mathbf{u}} \triangleq g \cdot \mathbf{u}$, with $\tilde{\mathbf{u}}(\tilde{\mathbf{x}}) = \mathbf{u}(\mathbf{x})$.

A differential invariant is a quantity that remains unchanged under the prolonged group action. The definition is given below.

**Definition 2** *Given a group $G$ acting on $X$, a $d$-th order **differential invariant** is a function $\mathcal{I} : X \times U^{(d)} \to \mathbb{R}$ such that*

$$\mathcal{I}(g \cdot (\mathbf{x}, \mathbf{u}^{(d)})) = \mathcal{I}(\mathbf{x}, \mathbf{u}^{(d)}), \quad \forall g \in G, (\mathbf{x}, \mathbf{u}^{(d)}) \in X \times U^{(d)}, \tag{2}$$

*where $g \cdot (\mathbf{x}, \mathbf{u}^{(d)})$ denotes the prolonged group action on the jet space $X \times U^{(d)}$.*

The definition can be extended to the multi-dimensional case. Specifically, we call $\boldsymbol{\mathcal{I}} = (\mathcal{I}_1, \ldots, \mathcal{I}_k)^\top$ an $k$**-dimensional differential invariant**. In addition, we define **relative differential invariants**, which may transform with a weight function under the group action:

$$\mathcal{R}(g \cdot (\mathbf{x}, \mathbf{u}^{(d)})) = w(g, \mathbf{x}) \cdot \mathcal{R}(\mathbf{x}, \mathbf{u}^{(d)}), \tag{3}$$

where $w(g, \mathbf{x})$ is a scalar weight depending on the group element $g$ and the point $\mathbf{x}$. Notably, differential invariants are closely tied to equivariance, as a (multi-dimensional) differential invariant $\boldsymbol{\mathcal{I}}$ yields an equivariant operator $\hat{\boldsymbol{\mathcal{I}}}(\mathbf{u})(\mathbf{x}) \triangleq \boldsymbol{\mathcal{I}}(\mathbf{x}, \mathbf{u}^{(d)})$ satisfying $\hat{\boldsymbol{\mathcal{I}}}(g \cdot \mathbf{u}) = g \cdot \hat{\boldsymbol{\mathcal{I}}}(\mathbf{u})$.

In this work, we focus on constructing such differential invariants and using them to build projective equivariant operators for neural networks.

## 2.2 Method of moving frames

The method of moving frames is a powerful technique for deriving differential invariants [Olver, 2003, 2015]. We begin with the definition of a moving frame.

**Definition 3** *[Olver, 2015] Let $G$ be a Lie group acting on a manifold $\mathcal{M}$. A **moving frame** is a map $\eta : \mathcal{M} \to G$ such that*

$$\eta(g \cdot z) = \eta(z) \cdot g^{-1}, \quad g \in G, z \in \mathcal{M}. \tag{4}$$

Given a moving frame, the **invariantization** of a function $F : \mathcal{M} \to \mathbb{R}$ is defined as

$$\iota(F)(z) \triangleq F(\eta(z) \cdot z), \tag{5}$$

which converts an arbitrary function $F$ into a group-invariant function satisfying $\iota(F)(g \cdot z) = \iota(F)(z)$. More generally, we can define an invariant as $\mathcal{I}(g \cdot z) = \mathcal{I}(z), g \in G, z \in \mathcal{M}$. In our context, the manifold of interest is the jet space $\mathcal{M} = X \times U^{(n)}$ and we focus on differential invariants.

A necessary and sufficient condition for the existence of a moving frame is that the group $G$ acts freely and regularly on the manifold $\mathcal{M}$. Under this condition, a moving frame can be constructed via a cross section, as described below:

**Theorem 4** *[Olver, 2015] Let $G$ be a $r$-dimensional Lie group acting freely and regularly on a $m$-dimensional manifold $\mathcal{M}$. Given local coordinates $z = (z_1, \ldots, z_m)$ on $\mathcal{M}$, let $\mathcal{K}$ be a cross section of the form $\mathcal{K} = \{z_1 = c_1, z_2 = c_2, \ldots, z_r = c_r\} \subset \mathcal{M}$, where $c_i$ are constants. Then for $z \in \mathcal{M}$, there exists a unique $g \in G$ such that $g \cdot z \in K$. Defining $\eta(z) = g$, namely $\eta(z) \cdot z \in \mathcal{K}$, yields a map $\eta : \mathcal{M} \to G$, which is a moving frame.*

Here, the group action is said to be **free** if for any $g \in G$, $g \cdot z = z$ implies $g = e$, where $e$ is the identity element of the group. Usually, the group action can be made free by increasing the order of the jet space. The action is **regular** if the orbits form a regular foliation, which is typically satisfied in common groups.

With a moving frame obtained from Theorem 4, we can construct a complete set of fundamental invariants, meaning any invariant can be expressed as a combination of these fundamental invariants.

**Theorem 5** *[Olver, 2015] Let $\eta : \mathcal{M} \to G$ be a moving frame from Theorem 4 and define $w(g, z) \triangleq g \cdot z$. Then*

$$w(\eta(z), z) = (c_1, c_2, \ldots, c_r, w_{r+1}(\eta(z), z), \ldots, w_m(\eta(z), z)), \tag{6}$$

*where $\mathcal{I}_1(z) \triangleq w_{r+1}(\eta(z), z)$, ..., $\mathcal{I}_{m-r}(z) \triangleq w_m(\eta(z), z)$ constitute a complete system of functionally independent invariants, called **fundamental invariants**.*

This theorem provides a method to construct fundamental invariants via the moving frame and indicates that the number of fundamental invariants is $m - r$. In the following sections, we will leverage these results to derive projective differential invariants.

### 2.3 Projective transformation

Projective transformations are ubiquitous in the visual world as two different views of the same planar object can be related by a 2D projective transformation. A standard projective group action is described by the projective special linear group $\mathrm{PSL}(3, \mathbb{R})$ acting on the 2D projective plane $\mathbb{RP}^2$, which can be interpreted as the set of equivalence classes of points $(x, y, p) \sim (cx, cy, cp)$ for any $c \neq 0$. Points in $\mathbb{RP}^2$ with $p \neq 0$ can be represented in inhomogeneous coordinates as $(x, y)$, corresponding to the homogeneous coordinate $(x, y, 1)$. Thus, the action of a projective transformation on 2D coordinates can be written as

$$\tilde{x} = \frac{\alpha x + \beta y + \gamma}{\rho x + \sigma y + \tau}, \tilde{y} = \frac{\lambda x + \mu y + \nu}{\rho x + \sigma y + \tau}, \tag{7}$$

where the transformation is parameterized by the coefficient matrix

$$\mathbf{P} = \begin{pmatrix} \alpha & \beta & \gamma \\ \lambda & \mu & \nu \\ \rho & \sigma & \tau \end{pmatrix}. \tag{8}$$

Since the transformation is defined up to a nonzero scaling factor, we can normalize by requiring the determinant of $\mathbf{P}$ to be 1, i.e., $\Delta = \det(\mathbf{P}) = 1$. Thus, there are 8 independent degrees of freedom. The transformation reduces to an affine transformation when $\rho = \sigma = 0$, while a pure projective transformation, characterized by $\rho^2 + \sigma^2 \neq 0$, exhibits nonlinear behavior. Thus, projective transformations represent a more general and complex class of geometric transformations.

For an $n$-dimensional function $\mathbf{u}(x, y)$, the projective transformation of coordinates induces a natural action on the function, $\tilde{\mathbf{u}}(\tilde{x}, \tilde{y}) = \mathbf{u}(x, y)$, which can be further prolonged to its derivatives. We denote the derivatives of the $i$-th component function $u^{[i]}$ as

$$u_{jk}^{[i]} \triangleq D_x^j D_y^k u^{[i]}, \tag{9}$$

where $D_x$ and $D_y$ are the differentiation operators with respect to $x$ and $y$, respectively. Under a projective transformation, these derivatives transform as

$$u_{jk}^{[i]} \mapsto \tilde{u}_{jk}^{[i]} = D_{\tilde{x}}^j D_{\tilde{y}}^k \tilde{u}^{[i]}, \tag{10}$$

with the transformed differential operators given by

$$D_{\tilde{x}} = \frac{\rho x + \sigma y + \tau}{\Delta} \left( ((\mu\rho - \lambda\sigma)x + \mu\tau - \nu\sigma)D_x + ((\mu\rho - \lambda\sigma)y - \lambda\tau + \nu\rho)D_y \right), \tag{11}$$

$$D_{\tilde{y}} = \frac{\rho x + \sigma y + \tau}{\Delta} \left( ((\alpha\sigma - \beta\rho)x - \beta\tau + \gamma\sigma)D_x + ((\alpha\sigma - \beta\rho)y + \alpha\tau - \gamma\rho)D_y \right). \tag{12}$$

## 2.4 Projective differential invariants of multi-dimensional functions

The projective group action is not free on the second-order jet space for scalar functions, requiring prolongation to the third-order jet space to achieve freeness. This leads to complex formulations [Olver, 2023], which may limit the practicality of the resulting invariants due to their complexity and computational cost. Moreover, in practice, third-order derivatives are harder to estimate reliably from data than lower-order ones. In this work, we focus on multi-dimensional functions, which naturally align with applications such as color image processing. In this setting, the group action is free on the second-order jet space, allowing the existence of second-order differential invariants. Using the method of moving frames, we can derive these invariants, where the choice of cross section significantly influences the simplicity of the resulting expressions. Although the cross section proposed by Olver [2023] can be extended to the multi-dimensional case, the resulting invariants tend to be lengthy, typically involving hundreds of terms, which makes them less practical. Instead, we propose an alternative cross section that leverages multiple dimensions while relying only on derivatives up to second order, yielding invariants with significantly more concise and tractable forms.

Specifically, we choose the following cross section:

$$\mathcal{K} = \{x = y = 0, u_x^{[1]} = 1, u_y^{[1]} = 0, u_x^{[2]} = 0, u_y^{[2]} = 1, u_{xx}^{[1]} = u_{xy}^{[1]} = 0\}. \tag{13}$$

Hereafter, we use the standard shorthand notation for partial derivatives, e.g., $u_x, u_y, u_{xx}, u_{xy}, u_{yy}$. The above cross section defines 8 normalization equations, which, together with the constraint $\Delta = 1$, determine all group parameters, thereby establishing the moving frame $\eta$. The detailed derivation of the moving frame is provided in the Appendix.

With the moving frame $\eta$ constructed, we can then apply the invariantization process according to Theorem 5 to obtain a complete set of fundamental differential invariants. For the coordinates involved in the cross section, we have

$$\iota(x) = 0, \iota(y) = 0, \iota(u_x^{[1]}) = 1, \iota(u_y^{[1]}) = 0,$$
$$\iota(u_x^{[2]}) = 0, \iota(u_y^{[2]}) = 1, \iota(u_{xx}^{[1]}) = 0, \iota(u_{xy}^{[1]}) = 0.$$

The remaining coordinates of the second-order jet space yield the following differential invariants:

$$\iota(u_{yy}^{[1]}) = \frac{\mathcal{T}_{111}}{\mathcal{J}_{12}^2}, \tag{14}$$

$$\iota(u_{xx}^{[2]}) = \frac{\mathcal{T}_{222}}{\mathcal{J}_{12}^2}, \tag{15}$$

$$\iota(u_{xy}^{[2]}) = -\frac{\mathcal{T}_{212} + 2\mathcal{T}_{122}}{2\mathcal{J}_{12}^2}, \tag{16}$$

$$\iota(u_{yy}^{[2]}) = \frac{\mathcal{T}_{121} + 2\mathcal{T}_{112}}{\mathcal{J}_{12}^2}, \tag{17}$$

$$\iota(u_x^{[i]}) = -\frac{\mathcal{J}_{2i}}{\mathcal{J}_{12}}, \quad 3 \le i \le n, \tag{18}$$

$$\iota(u_y^{[i]}) = \frac{\mathcal{J}_{1i}}{\mathcal{J}_{12}}, \quad 3 \le i \le n, \tag{19}$$

$$\iota(u_{xx}^{[i]}) = \frac{\mathcal{J}_{12}\mathcal{T}_{2i2} + \mathcal{J}_{2i}\mathcal{T}_{212}}{\mathcal{J}_{12}^3}, \quad 3 \le i \le n, \tag{20}$$

$$\iota(u_{xy}^{[i]}) = -\frac{2\mathcal{J}_{12}\mathcal{T}_{1i2} + 2\mathcal{J}_{12}\mathcal{T}_{21i} + 3\mathcal{J}_{2i}\mathcal{T}_{112}}{\mathcal{J}_{12}^3}, \quad 3 \le i \le n, \tag{21}$$

$$\iota(u_{yy}^{[i]}) = \frac{\mathcal{J}_{12}\mathcal{T}_{1i1} + 2\mathcal{J}_{1i}\mathcal{T}_{112}}{\mathcal{J}_{12}^3}, \quad 3 \le i \le n, \tag{22}$$

where $\mathcal{J}_{ij}$ and $\mathcal{T}_{ijk}$ are two key quantities defined as:

$$\mathcal{J}_{ij} \triangleq u_x^{[i]} u_y^{[j]} - u_x^{[j]} u_y^{[i]}, \tag{23}$$

$$\mathcal{T}_{ijk} \triangleq u_{xx}^{[i]} u_y^{[j]} u_y^{[k]} + u_{yy}^{[i]} u_x^{[j]} u_x^{[k]} - u_{xy}^{[i]} (u_x^{[j]} u_y^{[k]} + u_x^{[k]} u_y^{[j]}), \tag{24}$$

satisfying $\mathcal{J}_{ii} = 0$, $\mathcal{J}_{ij} = -\mathcal{J}_{ji}$, and $\mathcal{T}_{ijk} = \mathcal{T}_{kji}$.

The invariants (14)-(22), together with the obvious zeroth-order invariants

$$S_0 = \{u^{[i]} \mid 1 \le i \le n\},$$

form a complete set of second-order fundamental differential invariants of the projective group. Compared to projective invariants for scalar functions [Olver, 2023], our results involve up to second-order derivatives and are expressed in a more concise form.

## 2.5 Fundamental components of projective differential invariants

In the previous subsection, we have derived a complete set of second-order fundamental differential invariants. While relatively concise, their expressions are asymmetric and depend on the specific choice of the first two dimensions used in the cross section. To obtain a simpler, more unified, and elegant formulation, we conduct a deeper analysis of the fundamental components of these invariants. This enables us to further simplify their structure while preserving completeness.

Note that the numerators and denominators in (14)-(22) are all relative invariants. Thus, we focus on the properties of these relative invariants, as absolute invariants can be obtained by taking the ratio of two relative invariants with the same weight. Moreover, since the expressions are built from the basic quantities $\mathcal{J}_{ij}$ and $\mathcal{T}_{ijk}$, we will delve into their transformation properties and algebraic relationships.

We first present three classes of simplified relative differential invariants of the projective group.

**Theorem 6** *Let $W = \frac{(\rho x + \sigma y + \tau)^3}{\Delta}$. Then the following quantities are relative differential invariants of the projective group:*

- *For $i \ne j$, $\mathcal{J}_{ij}$ is a relative differential invariant of weight $W$.*

- *For $1 \le i \le n$, $\mathcal{T}_{iii}$ is a relative differential invariant of weight $W^2$.*

- *For $1 \le i, j \le n$, $\mathcal{T}_{iji} + 2\mathcal{T}_{iij}$ is a relative differential invariant of weight $W^2$.*

These relative invariants are not functionally independent; rather, they can be transformed into one another. Given that there are $6n - 6$ second-order fundamental differential invariants according to Section 2.4, we expect a complete and independent set of relative invariants to contain $6n - 5$ elements. To this end, we investigate the transformation rules among the relative invariants and aim to identify a minimal generating set sufficient to express all fundamental differential invariants. We start with the transformation properties of $\mathcal{J}_{ij}$.

**Theorem 7** *For any indices $1 \le i_1, i_2, i_3, i_4 \le n$, the following equation holds:*

$$\mathcal{J}_{i_1 i_2} \cdot \mathcal{J}_{i_3 i_4} + \mathcal{J}_{i_1 i_3} \cdot \mathcal{J}_{i_4 i_2} + \mathcal{J}_{i_1 i_4} \cdot \mathcal{J}_{i_2 i_3} = 0. \tag{25}$$

This implies that for any four distinct indices $i_1, i_2, i_3, i_4$, the six pairwise combinations of $\mathcal{J}_{ij}$ are dependent such that once any five are known, the remaining one can be determined. Based on Theorem 7, we can construct a subset of $\{\mathcal{J}_{ij} \mid i \ne j\}$ that is sufficient to express all $\mathcal{J}_{ij}$.

**Theorem 8** *Define the following sets of relative invariants:*

$$S_1 \triangleq \{\mathcal{J}_{12}, \mathcal{J}_{23}, \ldots, \mathcal{J}_{n-1,n}\}, \tag{26}$$

$$S_2 \triangleq \{\mathcal{J}_{13}, \mathcal{J}_{24}, \ldots, \mathcal{J}_{n-2,n}\}. \tag{27}$$

*Then $S_1 \cup S_2$ is a generating set for the collection $\{\mathcal{J}_{ij} \mid i \ne j\}$, meaning that any $\mathcal{J}_{ij}$ can be expressed as a functional combination of these elements.*

According to Theorem 7, $\mathcal{J}_{i,i+3}$ can be written in terms of $\mathcal{J}_{i,i+1}, \mathcal{J}_{i+1,i+2}, \mathcal{J}_{i+2,i+3}, \mathcal{J}_{i,i+2}$, and $\mathcal{J}_{i+1,i+3}$, all of which belong to $S_1 \cup S_2$. By induction, any $\mathcal{J}_{ij}$ can thus be recovered from the generating set. A complete and rigorous proof is provided in the Appendix.

Before establishing the transformation relationships for $\mathcal{T}_{ijk}$, we provide a more compact representation of $\mathcal{J}_{ij}$ and $\mathcal{T}_{ijk}$ to clarify their structural relationships:

$$\mathcal{J}_{ij} = \mathbf{g}_i^\top \mathbf{Q} \mathbf{g}_j, \tag{28}$$

$$\mathcal{T}_{ijk} = \mathbf{g}_i^\top \mathbf{Q}^\top \mathbf{H}_j \mathbf{Q} \mathbf{g}_k, \tag{29}$$

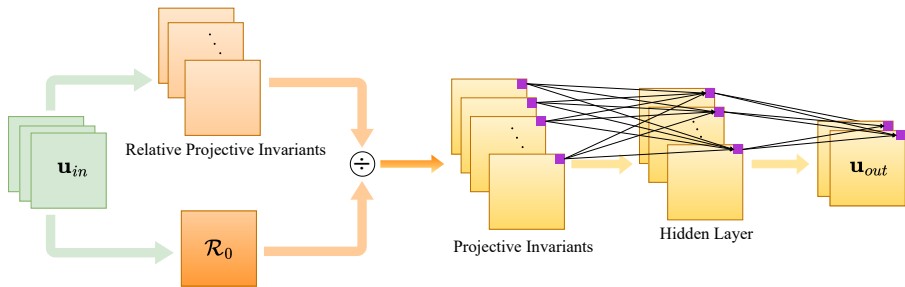

Figure 2: A single layer of PDINet. (1) Compute projective invariants from relative projective invariants. (2) Combine these invariants with an MLP to produce equivariant outputs.

where $\mathbf{g}_i \triangleq \left( u_x^{[i]}, u_y^{[i]} \right)^\top$ is the gradient of the $i$-th component function, $\mathbf{H}_j$ is the Hessian matrix of the $j$-th component function, and $\mathbf{Q}$ is a fixed orthogonal matrix defined as:

$$\mathbf{Q} \triangleq \left( \begin{array}{cc} 0 & 1 \\ -1 & 0 \end{array} \right).$$

Notably, $\{\mathcal{J}_{ij}\}$ can be used to establish the transformation relationships between the gradients.

**Lemma 9** *For any distinct indices $i, j, k$, the gradient $\mathbf{g}_k$ can be expressed in terms of $\mathbf{g}_i$ and $\mathbf{g}_j$ as:*

$$\mathbf{g}_k = \frac{\mathcal{J}_{kj}}{\mathcal{J}_{ij}}\mathbf{g}_i + \frac{\mathcal{J}_{ki}}{\mathcal{J}_{ji}}\mathbf{g}_j. \tag{30}$$

Using Lemma 9 and the form of $\mathcal{T}_{ijk}$ in (29), we derive the transformation rule for $\mathcal{T}_{ijk}$ given $\{\mathcal{J}_{ij}\}$.

**Theorem 10** *Let $i \neq k$, and let $i', j, k'$ be arbitrary indices. Then $\mathcal{T}_{i'jk'}$ can be expressed as:*

$$\mathcal{T}_{i'jk'} = \frac{\mathcal{J}_{i'k}\mathcal{J}_{k'k}}{\mathcal{J}_{ik}^2}\mathcal{T}_{iji} - \frac{\mathcal{J}_{i'k}\mathcal{J}_{k'i} + \mathcal{J}_{i'i}\mathcal{J}_{k'k}}{\mathcal{J}_{ik}^2}\mathcal{T}_{ijk} + \frac{\mathcal{J}_{i'i}\mathcal{J}_{k'i}}{\mathcal{J}_{ik}^2}\mathcal{T}_{kjk}. \tag{31}$$

This result implies that, for a fixed $j$, the triplet $\{\mathcal{T}_{iji}, \mathcal{T}_{ijk}, \mathcal{T}_{kjk}\}$ serves as a generating set for $\{\mathcal{T}_{i'jk'} \mid 1 \leq i', k' \leq n\}$, provided $\{\mathcal{J}_{ij}\}$ is known.

With the transformation relationships for $\mathcal{J}_{ij}$ and $\mathcal{T}_{ijk}$ established, we can now construct a minimal set of relative invariants that suffices to express all the relative invariants in Theorem 6.

**Theorem 11** *Define the following sets of relative invariants:*

$$S_3 \triangleq \{\mathcal{T}_{111}, \mathcal{T}_{222}, \ldots, \mathcal{T}_{nnn}\}, \tag{32}$$

$$S_4 \triangleq \{\mathcal{T}_{121} + 2\mathcal{T}_{112}, \mathcal{T}_{131} + 2\mathcal{T}_{113}, \ldots, \mathcal{T}_{n-1,n,n-1} + 2\mathcal{T}_{n-1,n-1,n}\}, \tag{33}$$

$$S_5 \triangleq \{\mathcal{T}_{212} + 2\mathcal{T}_{221}, \mathcal{T}_{323} + 2\mathcal{T}_{332}, \ldots, \mathcal{T}_{n,n-1,n} + 2\mathcal{T}_{n,n,n-1}\}. \tag{34}$$

*Then $S_1 \cup S_2 \cup S_3 \cup S_4 \cup S_5$ can express all the relative invariants in Theorem 6.*

It can be shown (see the Appendix) that the fundamental differential invariants (14)-(22) can be expressed via the relative invariants in Theorem 6. Therefore, we arrive at the following conclusion:

**Theorem 12** *The union $S \triangleq S_0 \cup S_1 \cup S_2 \cup S_3 \cup S_4 \cup S_5$ forms a complete set of relative differential invariants to express all second-order differential invariants of the projective group.*

This set contains exactly $6n - 5$ elements, matching the expected minimal number needed to express all second-order fundamental differential invariants. Compared to the invariants derived in Section 2.4, the current formulation is further simplified and independent of the specific choice of the first two dimensions in the cross section, exhibiting a more unified structure. Moreover, while the original invariants involve polynomials of degree up to five, the present set only contains at most cubic expressions, resulting in lower computational complexity.

## 2.6 Projective equivariant networks

As discussed before, with a set of relative invariants obtained, we can convert them into absolute invariants by dividing each by another relative invariant with the same weight. We apply this procedure to relative invariants in $S$ to construct a complete set of fundamental differential invariants.

Specifically, we select $\mathcal{R}_0 = \frac{1}{n}(\mathcal{J}_{12} + \mathcal{J}_{23} + \ldots + \mathcal{J}_{n1})$ as the denominator, which is a relative invariant of weight $W$. We keep the elements in $S_0$ unchanged, divide the elements in $S_1 \cup S_2$ by $\mathcal{R}_0$, and divide those in $S_3 \cup S_4 \cup S_5$ by $\mathcal{R}_0^2$. This yields a set of differential invariants sufficient to express all second-order fundamental differential invariants. In fact, this set with $6n - 5$ invariants may contain one redundant element, but completeness is of greater concern. To avoid division by zero, we add a positive constant $\epsilon$ to the denominator during division, enhancing numerical stability.

Let $\boldsymbol{\mathcal{I}} = (\mathcal{I}_1, \ldots, \mathcal{I}_N)^\top$ denote the set of differential invariants we obtained, which naturally induces an equivariant operator $\hat{\boldsymbol{\mathcal{I}}}$. Theoretically, the invariants $\mathcal{I}_1, \ldots, \mathcal{I}_N$ are sufficient to express all second-order differential invariants. In practice, we leverage the expressive power of neural networks to combine $\mathcal{I}_1, \ldots, \mathcal{I}_N$ using a two-layer MLP to produce the output [Li et al., 2024, 2025]. This leads to a learnable equivariant operator:

$$\mathbf{u}_{out} = \mathbf{h}_\theta \circ \hat{\boldsymbol{\mathcal{I}}}(\mathbf{u}_{in}), \tag{35}$$

where $\mathbf{h}_\theta$ is an MLP parameterized by $\theta$. By integrating this operator into standard network architectures, we can build projective equivariant models. We refer to the resulting model as the **Projective Differential Invariant Network** (**PDINet**), as illustrated in Figure 2.

## 3 Experiments

For empirical evaluation, we conduct image classification tasks under out-of-distribution settings, where models are trained on the original dataset and tested on images deformed by projective transformations. We adopt ResNet-18 [He et al., 2016] as the backbone and replace its convolutional layers with our equivariant operators defined in (35) to construct a projective equivariant network, PDINet. As the main counterpart, we consider homConv, a projective equivariant model proposed by MacDonald et al. [2022]. To ensure a fair comparison, we attempted to implement homConv using the same backbone. However, homConv relies on group sampling, which leads to exponential memory growth with network depth, resulting in out-of-memory (OOM) issues. Therefore, we follow the original network configuration of MacDonald et al. [2022] and reduce the number of samples to avoid OOM. In addition, we also include ResNet-18 trained with projective data augmentation (DA) as a reference baseline.

### 3.1 Proj-STL-10

STL-10 [Coates et al., 2011] is a dataset containing 5000 training images and 8000 test images. Each image has a resolution of $96 \times 96$ with RGB channels. We apply random projective transformations to the test set to generate the Proj-STL-10 dataset. Models are trained on the original STL-10 dataset (or with projective data augmentation for the DA baseline) and evaluated on Proj-STL-10, forming a challenging out-of-distribution setting that assesses the model's ability to generalize beyond the training distribution.

Table 1: Test accuracy (%) on Proj-STL-10.

| Model | Accuracy | # Params |
|---|---|---|
| ResNet-18 | $39.73_{\pm 0.33}$ | 11.18M |
| ResNet-18 with DA | $48.73_{\pm 0.53}$ | 11.18M |
| homConv | $20.88_{\pm 0.32}$ | 376K |
| PDINet (ours) | $\mathbf{51.12}_{\pm 0.47}$ | 8.90M |

Each experiment is repeated five times with different random seeds, and we report the average accuracy and standard deviation in Table 1. PDINet substantially outperforms the standard ResNet-18 on the transformed test set, achieving an $11.39\%$ improvement in accuracy. While projective data augmentation considerably enhances ResNet-18's robustness, PDINet still achieves the highest accuracy. In contrast, homConv performs poorly due to its restriction on network depth.

### 3.2 Proj-Imagenette

Imagenette is a ten-class subset of the ImageNet dataset [Deng et al., 2009], consisting of 9469 training images and 3925 test images. All images are adapted to a uniform resolution of $256 \times 256$ for model input. We apply random projective transformations to the test set to generate the Proj-Imagenette dataset, while keeping the training set unchanged. This setup simulates an out-of-distribution scenario and evaluates the model's ability to generalize to geometric transformations.

Table 2: Test accuracy (%) on Proj-Imagenette.

| Model | Accuracy | # Params |
|---|---|---|
| ResNet-18 | $65.64_{\pm 0.39}$ | 11.18M |
| ResNet-18 with DA | $70.77_{\pm 0.76}$ | 11.18M |
| homConv | $25.72_{\pm 0.52}$ | 376K |
| PDINet (ours) | $\mathbf{71.30}_{\pm 0.45}$ | 8.90M |

Each experiment is repeated five times with different random seeds, and we report the mean $\pm$ standard deviation of test accuracy in Table 2. The results demonstrate that PDINet retains strong performance under distribution shift, outperforming the standard ResNet-18 by $5.66\%$. It confirms the effectiveness of the projective equivariance of our model in enhancing out-of-distribution generalization. Although ResNet-18 with DA reduces the performance gap, it still lags behind PDINet. Meanwhile, the shallow homConv model struggles to handle higher-resolution inputs due to its limited depth.

Detailed experimental settings and implementation details can be found in the Appendix.

## 4 Conclusion

In this work, we propose **PDINet**, a framework for projective equivariant networks, based on second-order differential invariants of the projective group. Our method overcomes the exponential memory growth encountered by homConv [MacDonald et al., 2022], enabling effective scaling to deeper networks. Leveraging the moving frame method and a carefully chosen cross section tailored to multi-dimensional functions, we derive a complete and concise set of second-order projective fundamental differential invariants. Further analysis reveals transformation relationships among projective invariants, allowing us to obtain a unified and simplified formulation that enhances both theoretical clarity and computational efficiency. Building upon these invariants, we design a learnable projective equivariant operator that can be seamlessly integrated into various network architectures. It is the first time to achieve full projective equivariance in deep networks without group sampling or discretization. Experiments under out-of-distribution settings demonstrate the strong generalization ability of our model. With the prevalence and significance of projective transformations in vision, PDINet holds promising potential for broader applications in computer vision.

One limitation of our approach is that the second-order invariants we derive vanish in the one-dimensional case, preventing the direct application of PDINet to grayscale images. In addition, this work focuses on group actions on scalar fields and does not yet cover more general cases involving arbitrary group representations, which we consider a valuable direction for future research.

## Acknowledgments

Z. Lin was supported by National Key R&D Program of China (2022ZD0160300), the NSF China (No. 62276004) and the State Key Laboratory of General Artificial Intelligence.

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

# A Related work

## A.1 Differential invariants

Differential invariants are quantities involving derivatives that remain unchanged under the action of a given transformation group [Olver, 1993]. They have proven useful in various image analysis applications [Mundy and Zisserman, 1992, Olver et al., 1999, Hartley and Zisserman, 2003, Li et al., 2018]. Leveraging the connection between differential invariants and symmetric partial differential equations (PDEs) [Olver, 1993], Liu et al. [2010, 2013] constructed learnable PDEs as linear combinations of differential invariants, achieving shift and rotation equivariance. This connection has also been explored for symmetry-informed discovery of governing PDEs [Hu et al., 2025d]. The method of moving frames offers a systematic framework for deriving differential invariants given a transformation group [Fels and Olver, 1999, Olver, 2003, 2015]. Based on this approach, Olver [2023] provided a characterization of differential invariants of the projective group. However, since the projective group does not act freely on the second-order jet space of scalar functions, Olver [2023] prolonged the group action to the third-order jet space, resulting in complex expressions that are computationally expensive and difficult to estimate from data due to the involvement of higher-order derivatives. Differently, our work focuses on multi-dimensional functions, where the projective group acts freely on the second-order jet space. This allows us to derive a complete and concise set of second-order fundamental differential invariants, which we further simplify through structural analysis.

## A.2 Equivariant networks

Early developments in equivariant networks primarily focused on designing architectures that are equivariant to discrete groups, such as cyclic or dihedral groups. A representative approach is the G-CNN [Cohen and Welling, 2016a] framework, which models feature maps as functions defined on a group and implements convolution-like operations via discrete permutations over the input domain. Building upon this viewpoint, subsequent works [Shen et al., 2020, Romero and Cordonnier, 2021, He et al., 2021] developed a broader range of group-equivariant operators beyond standard group convolutions. To achieve equivariance with respect to continuous groups, steerable CNNs [Cohen and Welling, 2016b, Weiler and Cesa, 2019] were proposed, which treat feature maps as fields transforming according to specified group representations. This framework enables principled handling of Euclidean groups in 2D and 3D settings [Weiler and Cesa, 2019, Fuchs et al., 2020, Wang et al., 2020, Zhao et al., 2022, Shen et al., 2022, Liao and Smidt, 2022, Hu et al., 2024, 2025a,b], by explicitly encoding the group action on the feature spaces.

Symmetries are ubiquitous and can be leveraged in model design, either from prior knowledge or discovered from data [Yang et al., 2023, 2024, Hu et al., 2025c,e]. However, generalizing equivariant architectures to high-dimensional Lie groups, such as the projective group, poses significant challenges. For such groups, deriving explicit steerable basis filters is often intractable, limiting the applicability of steerable CNNs, while G-CNN based methods require group discretization or sampling, introducing scalability issues. Bökman et al. [2023] explored equivariance in a projective sense and achieved equivariant models with respect to projective representations of certain simple groups, rather than the full projective group. Another related effort is Möbius Convolution (MC) [Mitchel et al., 2022], which achieves equivariance to spherical Möbius transformations for geometry and spherical image processing tasks. While both MC and our method aim to build equivariant models under certain projective group actions, MC operates on the Riemann sphere under the complex-valued $\mathrm{SL}(2, \mathbb{C})$, whereas our work considers planar projective transformations governed by $\mathrm{SL}(3, \mathbb{R})$, corresponding to the commonly studied projective transformations in vision. Additionally, EMLP [Finzi et al., 2021] and EKAN [Hu et al., 2025f] incorporate arbitrary matrix group equivariance into MLPs and KANs [Liu et al., 2025], respectively, but they target linear transformations on vectors or tensors and are not directly suitable for image inputs. Shen et al. [2024] combined rotation-equivariant networks with data augmentation to obtain nearly affine invariant features, without achieving full equivariance. LieConv [Finzi et al., 2020] attempts to handle Lie group equivariance by sampling from the Haar measure, but struggles to extend to complex groups due to inaccessibility to the Haar measure. To mitigate this problem, MacDonald et al. [2022] performed group convolutions by computing the integral on the Lie algebra, resulting in a projective equivariant model, homConv. Nevertheless, it suffers from exponential memory growth as network depth increases, making it impractical for deep networks. Mironenco and Forré [2024] improved sampling efficiency via group decompositions, but

their method remains constrained to affine subgroups. Besides these two mainstream approaches, G-CNNs and steerable CNNs, Li et al. [2024, 2025] resorted to another route by constructing equivariant networks based on differential invariants, achieving affine equivariance without requiring group sampling or discretization. However, their method is confined to affine groups and does not generalize to the projective group. In particular, projective differential invariants for multi-channel inputs have not yet been developed, and the SupNorm normalization technique used to construct affine invariants cannot be directly extended to the projective group. In our work, we target projective equivariance by deriving a complete set of second-order fundamental differential invariants for multi-channel inputs using a tailored moving frame construction. Through algebraic analysis, we obtain simplified projective invariants for practical use, which enable the construction of deep projective equivariant networks without relying on sampling or discretizing the group.

# B Detailed proofs

In this section, we provide the detailed derivation of the moving frame and complete proofs of Theorems 8, 11, and 12.

For completeness, we briefly comment on the results that can be verified primarily through direct computation, while omitting the detailed algebraic steps:

- Theorem 6 follows directly from the definitions of $\mathcal{J}_{ij}$ and $\mathcal{T}_{ijk}$, together with the transformation rules for first- and second-order derivatives under the group action derived from (11)-(12), followed by straightforward simplification.
- Theorem 7 and Lemma 9 can be verified by substituting the definition of $\mathcal{J}_{ij}$ and expanding the expressions algebraically.
- Theorem 10 can be proved by rewriting $\mathcal{T}_{i'jk'}$ in the compact form $\mathbf{g}_{i'}^{\top}\mathbf{Q}^{\top}\mathbf{H}_j\mathbf{Q}\mathbf{g}_{k'}$ as shown in (29), then applying Lemma 9 to express the gradient vectors $\mathbf{g}_{i'}$ and $\mathbf{g}_{k'}$ in terms of $\mathbf{g}_i$ and $\mathbf{g}_k$, and simplifying the resulting expression.

## B.1 Derivation of the moving frame

We begin by setting $\tilde{x} = \tilde{y} = 0$, yielding

$$\gamma = -\alpha x - \beta y, \tag{36}$$

$$\nu = -\lambda x - \mu y. \tag{37}$$

Next, imposing $\tilde{u}_{\tilde{x}}^{[1]} = 1, \tilde{u}_{\tilde{y}}^{[1]} = 0$ gives

$$\alpha = u_x^{[1]}(\rho x + \sigma y + \tau), \tag{38}$$

$$\beta = u_y^{[1]}(\rho x + \sigma y + \tau). \tag{39}$$

Similarly, enforcing $\tilde{u}_{\tilde{x}}^{[2]} = 0, \tilde{u}_{\tilde{y}}^{[2]} = 1$ leads to

$$\lambda = u_x^{[2]}(\rho x + \sigma y + \tau), \tag{40}$$

$$\mu = u_y^{[2]}(\rho x + \sigma y + \tau). \tag{41}$$

We then apply the conditions $\tilde{u}_{\tilde{x}\tilde{x}}^{[1]} = \tilde{u}_{\tilde{x}\tilde{y}}^{[1]} = 0$, which yield the equations for $\sigma$ and $\tau$.

$$\sigma = \rho \frac{\mathcal{J}_{12}(u_{yy}^{[1]}u_x^{[2]} - u_{xy}^{[1]}u_y^{[2]}) + u_y^{[2]}\mathcal{T}_{112}}{-u_x^{[1]}\mathcal{T}_{212} + 2u_x^{[2]}\mathcal{T}_{112}}, \tag{42}$$

$$\tau = \rho \frac{x(u_x^{[1]}\mathcal{T}_{212} - 2u_x^{[2]}\mathcal{T}_{112}) + y(u_y^{[1]}\mathcal{T}_{212} - 2u_y^{[2]}\mathcal{T}_{112})}{-u_x^{[1]}\mathcal{T}_{212} + 2u_x^{[2]}\mathcal{T}_{112}}. \tag{43}$$

Finally, setting $\Delta = 1$ determines the parameter $\rho$ as:

$$\rho = \frac{u_x^{[1]}\mathcal{T}_{212} - 2u_x^{[2]}\mathcal{T}_{112}}{2\mathcal{J}_{12}^{7/3}}. \tag{44}$$

With all the parameters now fully determined, we obtain the moving frame $\eta$.

## B.2 Proof of Theorem 8

*Proof.* Define $J_k \triangleq \{\mathcal{J}_{i,i+k} \mid 1 \leq i \leq n - k\}, k = 1, 2, \ldots, n - 1$. To prove the theorem, it suffices to show that each $J_k$ for $1 \leq k \leq n - 1$ can be expressed in terms of elements from $S_1 \cup S_2$.

We proceed by mathematical induction on $k$.

**Base case** ($k = 1, 2$): This holds by definition, as $J_1 = S_1$ and $J_2 = S_2$.

**Inductive step**: Assume that for all $k \leq l - 1$ with $l \geq 3$, each set $J_k$ can be expressed in terms of elements from $S_1 \cup S_2$. We aim to show that $J_l$ can also be written using elements from $S_1 \cup S_2$.

By the induction hypothesis, the following quantities can all be expressed by elements in $S_1 \cup S_2$:

$$\mathcal{J}_{i,i+1}, \quad \mathcal{J}_{i,i+l-1}, \quad \mathcal{J}_{i+1,i+l-1}, \quad \mathcal{J}_{i+1,i+l}, \quad \mathcal{J}_{i+l-1,i+l}.$$

For any $\mathcal{J}_{i,i+l} \in J_l$, note that when $l \geq 3$, the indices $i, i + 1, i + l - 1, i + l$ are distinct. By Theorem 7, it follows that $\mathcal{J}_{i,i+l}$ can be expressed as a function of these five relative invariants. Hence, $\mathcal{J}_{i,i+l}$ can be expressed by elements in $S_1 \cup S_2$.

By the principle of mathematical induction, all $J_k$ for any $1 \leq k \leq n - 1$ can be generated from $S_1 \cup S_2$, completing the proof. $\square$

## B.3 Proof of Theorem 11

*Proof.* Recall that $\mathcal{J}_{ij}$ has been established in Theorem 8, and $\mathcal{T}_{iii}$ is already included in $S_3$. So we focus on proving that $\mathcal{T}_{iji} + 2\mathcal{T}_{iij}$ can be expressed in terms of elements from $S_1 \cup S_2 \cup S_3 \cup S_4 \cup S_5$. Define

$$T_k^+ \triangleq \{\mathcal{T}_{i,i+k,i} + 2\mathcal{T}_{i,i,i+k} \mid 1 \leq i \leq n - k\}, \tag{45}$$

$$T_k^- \triangleq \{\mathcal{T}_{i,i-k,i} + 2\mathcal{T}_{i,i,i-k} \mid k + 1 \leq i \leq n\}. \tag{46}$$

Our goal is to show that each element in $T_k^+$ and $T_k^-$ for $1 \leq k \leq n - 1$ can be expressed in terms of elements from $S_1 \cup S_2 \cup S_3 \cup S_4 \cup S_5$. We present the proof for $T_k^+$; the case of $T_k^-$ follows analogously due to index symmetry.

We proceed by mathematical induction on $k$.

**Base case** ($k = 1$): This holds by definition, as $T_1^+ = S_4$.

**Inductive step**: Assume that for all $k \leq l - 1$ with $l \geq 2$, each element in $T_k^+$ can be expressed in terms of elements from $S_1 \cup S_2 \cup S_3 \cup S_4 \cup S_5$. We aim to show that every element $\mathcal{T}_{i,i+l,i} + 2\mathcal{T}_{i,i,i+l} \in T_l^+$ can also be expressed using this union. From the transformation rule in Theorem 10, we have:

$$\mathcal{T}_{i,i+l,i} + 2\mathcal{T}_{i,i,i+l} \tag{47}$$

$$= (C_1 \mathcal{T}_{i+l-1,i+l,i+l-1} - C_2 \mathcal{T}_{i+l-1,i+l,i+l} + C_3 \mathcal{T}_{i+l,i+l,i+l}) + 2(C_4 \mathcal{T}_{iii} - C_5 \mathcal{T}_{i,i,i+l-1}) \tag{48}$$

$$\begin{aligned} =& C_1(\mathcal{T}_{i+l-1,i+l,i+l-1} + 2\mathcal{T}_{i+l-1,i+l-1,i+l}) - \frac{1}{2}C_2(2\mathcal{T}_{i+l-1,i+l,i+l} + \mathcal{T}_{i+l,i+l-1,i+l}) \\ &+ C_3 \mathcal{T}_{i+l,i+l,i+l} + 2C_4 \mathcal{T}_{iii} - C_5(2\mathcal{T}_{i,i,i+l-1} + \mathcal{T}_{i,i+l-1,i}) \\ &- 2C_1 \mathcal{T}_{i+l-1,i+l-1,i+l} + \frac{1}{2}C_2 \mathcal{T}_{i+l,i+l-1,i+l} + C_5 \mathcal{T}_{i,i+l-1,i}, \end{aligned} \tag{49}$$

where the coefficients are defined as:

$$C_1 = \frac{\mathcal{J}_{i,i+l}^2}{\mathcal{J}_{i+l-1,i+l}^2}, \quad C_2 = \frac{2\mathcal{J}_{i,i+l}\mathcal{J}_{i,i+l-1}}{\mathcal{J}_{i+l-1,i+l}^2}, \quad C_3 = \frac{\mathcal{J}_{i,i+l-1}^2}{\mathcal{J}_{i+l-1,i+l}^2},$$

$$C_4 = \frac{\mathcal{J}_{i+l,i+l-1}}{\mathcal{J}_{i,i+l-1}}, \quad C_5 = \frac{\mathcal{J}_{i+l,i}}{\mathcal{J}_{i,i+l-1}}. \tag{50}$$

By the induction hypothesis, $2\mathcal{T}_{i,i,i+l-1} + \mathcal{T}_{i,i+l-1,i} \in T_{l-1}^+$ can be expressed using elements from $S_1 \cup S_2 \cup S_3 \cup S_4 \cup S_5$. In addition, it holds that $\mathcal{T}_{iii} \in S_3, \mathcal{T}_{i+l-1,i+l,i+l-1} + 2\mathcal{T}_{i+l-1,i+l-1,i+l} \in S_4$,

and $2\mathcal{T}_{i+l-1,i+l,i+l} + \mathcal{T}_{i+l,i+l-1,i+l} \in S_5$. The only remaining terms are:

$$-2C_1\mathcal{T}_{i+l-1,i+l-1,i+l} + \frac{1}{2}C_2\mathcal{T}_{i+l,i+l-1,i+l} + C_5\mathcal{T}_{i,i+l-1,i} \tag{51}$$

$$= -2C_1\mathcal{T}_{i+l-1,i+l-1,i+l} + \frac{1}{2}C_2\mathcal{T}_{i+l,i+l-1,i+l}$$
$$+ C_5(C_1\mathcal{T}_{i+l-1,i+l-1,i+l-1} - C_2\mathcal{T}_{i+l-1,i+l-1,i+l} + C_3\mathcal{T}_{i+l,i+l-1,i+l}) \tag{52}$$
$$= C_5C_1\mathcal{T}_{i+l-1,i+l-1,i+l-1}, \tag{53}$$

which can be expressed in terms of $S_1 \cup S_2 \cup S_3 \cup S_4 \cup S_5$ as well. Hence, the full expression for $\mathcal{T}_{i,i+l,i} + 2\mathcal{T}_{i,i,i+l}$ is a combination of elements in $S_1 \cup S_2 \cup S_3 \cup S_4 \cup S_5$.

Therefore, by the principle of mathematical induction, all elements in $T_k^+$ for any $1 \leq k \leq n-1$ can be generated from the given sets. $\qquad\square$

### B.4 Proof of Theorem 12

*Proof.* To prove Theorem 12, it suffices to show that all second-order fundamental differential invariants of the projective group, namely the quantities in (14)-(22), along with the set $S_0$, can be expressed in terms of the elements in $S$. Since $S_0 \subset S$, we may disregard $S_0$ in the following discussion.

Moreover, Theorem 11 has established that all relative invariants given in Theorem 6 can be generated by elements in the union $S_1 \cup S_2 \cup S_3 \cup S_4 \cup S_5 \subset S$. Therefore, it remains to prove that the expressions (14)-(22) can be written in terms of the relative invariants from Theorem 6.

Among these fundamental invariants, (14)-(19) can be directly expressed using the relative invariants from Theorem 6. Thus, it remains to focus on the final three expressions (20)-(22). Since the denominators are already included in the set $S$, it suffices to consider only the numerators:

$$\mathcal{J}_{12}\mathcal{T}_{2i2} + \mathcal{J}_{2i}\mathcal{T}_{212}, \quad 3 \leq i \leq n, \tag{54}$$
$$2\mathcal{J}_{12}\mathcal{T}_{1i2} + \mathcal{J}_{12}\mathcal{T}_{21i} + 3\mathcal{J}_{2i}\mathcal{T}_{112}, \quad 3 \leq i \leq n, \tag{55}$$
$$\mathcal{J}_{12}\mathcal{T}_{1i1} + 2\mathcal{J}_{1i}\mathcal{T}_{112}, \quad 3 \leq i \leq n. \tag{56}$$

We now show that each of these numerators can indeed be expressed using the relative invariants in Theorem 6.

For (54), we have

$$\mathcal{J}_{12}\mathcal{T}_{2i2} + \mathcal{J}_{2i}\mathcal{T}_{212} \tag{57}$$
$$= \mathcal{J}_{12}(\mathcal{T}_{2i2} + 2\mathcal{T}_{22i}) + \mathcal{J}_{2i}(\mathcal{T}_{212} + 2\mathcal{T}_{221}) - 2\mathcal{J}_{12}\mathcal{T}_{22i} - 2\mathcal{J}_{2i}\mathcal{T}_{221} \tag{58}$$
$$= \mathcal{J}_{12}(\mathcal{T}_{2i2} + 2\mathcal{T}_{22i}) + \mathcal{J}_{2i}(\mathcal{T}_{212} + 2\mathcal{T}_{221}) - 2\mathcal{J}_{12}(C_6\mathcal{T}_{221} + C_7\mathcal{T}_{222}) - 2\mathcal{J}_{2i}\mathcal{T}_{221} \tag{59}$$
$$= \mathcal{J}_{12}(\mathcal{T}_{2i2} + 2\mathcal{T}_{22i}) + \mathcal{J}_{2i}(\mathcal{T}_{212} + 2\mathcal{T}_{221}) - 2\mathcal{J}_{12}C_7\mathcal{T}_{222}, \tag{60}$$

where the coefficients are defined as:

$$C_6 = \frac{\mathcal{J}_{i2}}{\mathcal{J}_{12}}, \quad C_7 = \frac{\mathcal{J}_{i1}}{\mathcal{J}_{21}}. \tag{61}$$

For (55), we first rewrite $\mathcal{T}_{1i2}$ as:

$$\mathcal{T}_{1i2} = C_8\mathcal{T}_{1i1} + C_9\mathcal{T}_{1ii} \tag{62}$$

$$= C_8(\mathcal{T}_{1i1} + 2\mathcal{T}_{11i}) + \frac{1}{2}C_9(2\mathcal{T}_{1ii} + \mathcal{T}_{i1i}) - 2C_8\mathcal{T}_{11i} - \frac{1}{2}C_9\mathcal{T}_{i1i} \tag{63}$$

$$= C_8(\mathcal{T}_{1i1} + 2\mathcal{T}_{11i}) + \frac{1}{2}C_9(2\mathcal{T}_{1ii} + \mathcal{T}_{i1i})$$
$$- 2C_8(C_6\mathcal{T}_{111} + C_7\mathcal{T}_{112}) - \frac{1}{2}C_9(C_6^2\mathcal{T}_{111} + 2C_6C_7\mathcal{T}_{112} + C_7^2\mathcal{T}_{212}), \tag{64}$$

where the coefficients are defined as:

$$C_8 = \frac{\mathcal{J}_{2i}}{\mathcal{J}_{1i}} \quad C_9 = \frac{\mathcal{J}_{21}}{\mathcal{J}_{i1}}. \tag{65}$$

IV

Similarly, we rewrite $\mathcal{T}_{21i}$ as:

$$\mathcal{T}_{21i} = C_6\mathcal{T}_{211} + C_7\mathcal{T}_{212}. \tag{66}$$

Substituting these into (55), we have:

$$2\mathcal{J}_{12}\mathcal{T}_{1i2} + \mathcal{J}_{12}\mathcal{T}_{21i} + 3\mathcal{J}_{2i}\mathcal{T}_{112} \tag{67}$$

$$=2\mathcal{J}_{12}\left( C_8(\mathcal{T}_{1i1} + 2\mathcal{T}_{11i}) + \frac{1}{2}C_9(2\mathcal{T}_{1ii} + \mathcal{T}_{i1i})\right)$$

$$+ 2\mathcal{J}_{12}\left( -2C_8(C_6\mathcal{T}_{111} + C_7\mathcal{T}_{112}) - \frac{1}{2}C_9(C_6^2\mathcal{T}_{111} + 2C_6C_7\mathcal{T}_{112} + C_7^2\mathcal{T}_{212})\right)$$

$$+ \mathcal{J}_{12}(C_6\mathcal{T}_{211} + C_7\mathcal{T}_{212}) + 3\mathcal{J}_{2i}\mathcal{T}_{112} \tag{68}$$

$$=2\mathcal{J}_{12}\left( C_8(\mathcal{T}_{1i1} + 2\mathcal{T}_{11i}) + \frac{1}{2}C_9(2\mathcal{T}_{1ii} + \mathcal{T}_{i1i})\right) - \left(4\mathcal{J}_{12}C_6C_8 + \mathcal{J}_{12}C_9C_6^2\right)\mathcal{T}_{111}. \tag{69}$$

For (56), we have:

$$\mathcal{J}_{12}\mathcal{T}_{1i1} + 2\mathcal{J}_{1i}\mathcal{T}_{112} \tag{70}$$

$$=\mathcal{J}_{12}(\mathcal{T}_{1i1} + 2\mathcal{T}_{11i}) + 2\mathcal{J}_{1i}\mathcal{T}_{112} - 2\mathcal{J}_{12}\mathcal{T}_{11i} \tag{71}$$

$$=\mathcal{J}_{12}(\mathcal{T}_{1i1} + 2\mathcal{T}_{11i}) + 2\mathcal{J}_{1i}\mathcal{T}_{112} - 2\mathcal{J}_{12}(C_6\mathcal{T}_{111} + C_7\mathcal{T}_{112}) \tag{72}$$

$$=\mathcal{J}_{12}(\mathcal{T}_{1i1} + 2\mathcal{T}_{11i}) - 2\mathcal{J}_{12}C_6\mathcal{T}_{111} \tag{73}$$

This completes the proof that all second-order fundamental differential invariants in (14)-(22) are expressible in terms of the relative invariants in Theorem 6, and thus can be generated by $S$. $\qquad\square$

## C   Implementation details of PDINet

While our theoretical foundation is developed in the continuous setting, practical applications involve discrete image data defined on grid points, where derivatives must be approximated numerically. To this end, we estimate spatial derivatives using Gaussian derivatives [Li et al., 2018, He et al., 2022, Li et al., 2024]. For example, the partial derivative with respect to $x$ is computed via convolution as $\frac{\partial f}{\partial x} \approx f * \frac{\partial G_\sigma}{\partial x}$, where $G_\sigma$ is a Gaussian kernel with zero mean and standard deviation $\sigma$. In our implementation, we set $\sigma = 0.99$ and use a kernel size of 9.

As mentioned in Subsection 2.6, we construct projective invariants by dividing each relative invariant by a designated relative invariant $\mathcal{R}_0$ or $\mathcal{R}_0^2$, which is computed from the input image. To prevent division by zero, we add a positive constant $\epsilon$ to the denominator. We set $\epsilon = 1$. After obtaining the invariants, we apply SupNorm normalization [Li et al., 2024, 2025], which preserves equivariance and is beneficial for training stability. Then we combine the invariants using a two-layer MLP, which is implemented as a sequence of two $1 \times 1$ convolutions with a ReLU activation function in between.

Given a standard convolutional network architecture, we construct a projective equivariant network by replacing each convolutional layer with our equivariant operator. If a convolutional layer has stride greater than 1, we insert an average pooling layer with a kernel size equal to the stride before the second $1 \times 1$ convolutional layer in the equivariant operator. Additionally, we also apply the same pooling when computing $\mathcal{R}_0$ to ensure resolution consistency between the numerator and denominator. Figure 3 shows the training loss curve on the STL-10 dataset, demonstrating stable optimization behavior of PDINet during training.

## D   Experimental details

All experiments are conducted on a single NVIDIA RTX 3090 GPU. Each experiment is repeated five times with independently generated test sets using random projective transformations, and we report the mean accuracy and standard deviation.

**Experiments on Proj-STL-10**. Models are trained on the 5000 samples of the STL-10 training set and tested on the 8000 samples of the Proj-STL-10 test set, which is generated by applying projective transformations to each sample in the STL-10 test set. Specifically, we decompose a projective transformation into an affine transformation followed by horizontal and vertical pure projections,

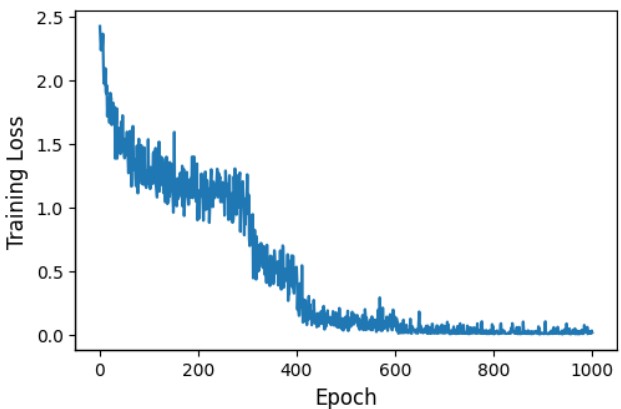

Figure 3: Training loss curve of PDINet on the STL-10 dataset.

defined as $(\tilde{x}, \tilde{y})^\top = (\frac{x}{1+c_1 x}, \frac{y}{1+c_1 x})^\top$ and $(\tilde{x}, \tilde{y})^\top = (\frac{x}{1+c_2 y}, \frac{y}{1+c_2 y})^\top$, respectively. The affine transformation consists of random rotation between $-90°$ and $90°$, scaling in the range $[0.9, 1.1]$, shear within $\pm 4°$, and translation within $[-0.1, 0.1]$, while the projection parameters $c_1$ and $c_2$ are uniformly sampled from $[-0.0001, 0.0001]$. All images are normalized by channel-wise mean subtraction and standard deviation division. Following [Sosnovik et al., 2019], data augmentation during training includes 12-pixel zero-padding followed by random cropping to $96 \times 96$, random horizontal flipping, and Cutout [DeVries and Taylor, 2017] with a single $32 \times 32$ hole. We train the models for 1000 epochs using SGD optimizer with Nesterov momentum of 0.9 and a batch size of 64. The initial learning rate is set to 0.1 and decayed by a factor of 0.2 at epochs 300, 400, 600, and 800. For the DA baseline, since the projective group is a complex non-compact group and the specific range or distribution of transformation parameters in test scenarios is typically unknown, we adopt a considerably wide range of geometric transformations. Specifically, we apply random rotation between $-180°$ and $180°$, scaling in the range $[0.3, 1.7]$, shear within $\pm 45°$, translation within $[-0.1, 0.1]$, and projection parameters $c_1, c_2$ uniformly sampled from $[-0.002, 0.002]$. For homConv, we use the Adam optimizer following the original setup of [MacDonald et al., 2022], as training with SGD fails to converge, while keeping other hyperparameters identical to those described above.

**Experiments on Proj-Imagenette.** Models are trained on the 9469 samples of the Imagenette training set and evaluated on the 3925 samples of the Proj-Imagenette test set. The Proj-Imagenette dataset is generated by applying projective transformations to each test image in the original Imagenette dataset, following the same procedure as Proj-STL-10. All images are normalized by subtracting the per-channel mean and dividing by the per-channel standard deviation. During training, data augmentation includes random resized cropping to $224 \times 224$ and random horizontal flipping. We train the models for 100 epochs using AdamW optimizer with a batch size of 64. The initial learning rate is set to 0.002 and decayed via a cosine annealing scheduler. The same strategy as in experiments on Proj-STL-10 is used for the DA baseline, while homConv is also trained with Adam as in [MacDonald et al., 2022].

## E  Additional experiments

### E.1  Equivariance error

Table 3: Equivariance error across different image resolutions.

| Image size | $16 \times 16$ | $32 \times 32$ | $64 \times 64$ | $128 \times 128$ | $256 \times 256$ |
|---|---|---|---|---|---|
| Equivariance error (%) | 0.34 | 0.14 | 0.08 | 0.04 | 0.02 |

Theoretically, the projective equivariance of our operators is rigorously guaranteed by the fundamental properties of differential invariants. In implementation, however, derivatives are estimated on discrete grids, which inevitably introduces minor equivariance errors. To quantitatively evaluate this effect,

we follow the protocol in [MacDonald et al., 2022] and define the equivariance error as

$$\text{Error} = \frac{\|g \cdot \psi(\mathbf{u}) - \psi(g \cdot \mathbf{u})\|^2}{\|g \cdot \psi(\mathbf{u})\|^2},$$

where $\psi$ denotes the equivariant layer and $g$ is a random projective transformation. Since the projective equivariance of PDINet is intrinsic and does not rely on training, we measure this error using a randomly initialized equivariant layer. We compute the error on the Imagenette test set, resizing images to multiple resolutions. As shown in Table 3, the equivariance error remains consistently small across all resolutions and decreases monotonically with increasing image size, which is expected due to more accurate derivative approximation at higher resolutions.

To further visualize equivariance, Figure 4 compares the features $g \cdot \psi(\mathbf{u})$ and $\psi(g \cdot \mathbf{u})$, showing that they are nearly identical and thus confirming the commutativity between the equivariant layer and projective transformations.

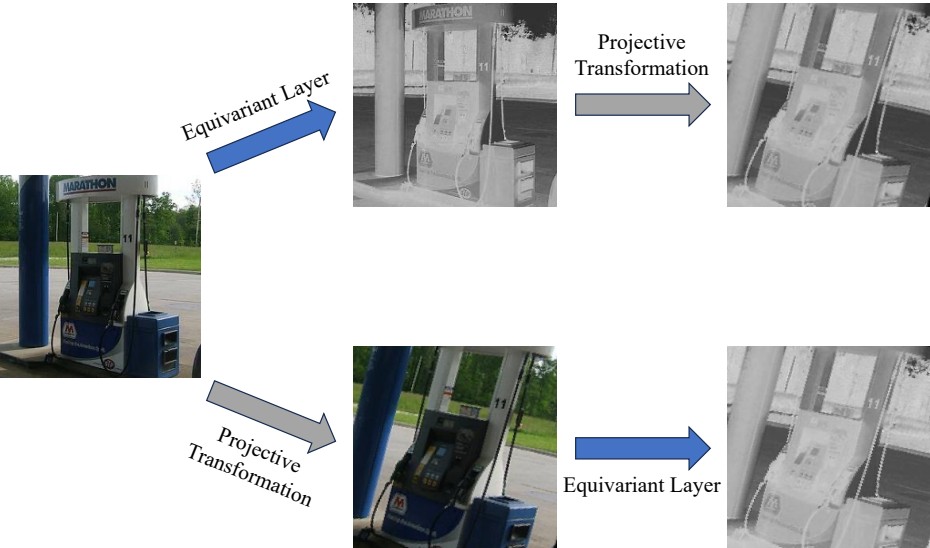

Figure 4: Visualization of projective equivariance: the features $g \cdot \psi(\mathbf{u})$ and $\psi(g \cdot \mathbf{u})$ remain consistent, illustrating the commutativity between the equivariant layer of PDINet and projective transformations.

### E.2   Computational complexity

Our projective equivariant layer exhibits linear growth in both time and space complexity with respect to the input size. Specifically, the computation includes estimating derivatives, computing differential invariants, and combining differential invariants, each with linear complexity. To quantify the overhead, we compare PDINet and ResNet-18 in terms of memory usage and FLOPs using *torchstat*, and report the results in Table 4. As shown, both models exhibit linear scaling with input resolution, highlighting the scalability of our model to higher-resolution inputs. While PDINet consumes slightly more memory than ResNet-18, it requires fewer FLOPs.

To provide an explicit runtime comparison, we further evaluate both models on the same hardware by measuring the total time required to process 1000 RGB images ($224 \times 224$) with a batch size of 100. Each experiment is repeated five times, and the mean and standard deviation are reported in Table 5. Despite its lower theoretical FLOPs, PDINet runs slower in practice, primarily due to the highly optimized low-level implementations of ResNet-18. We expect this gap can be narrowed through further engineering and implementation optimization.

### E.3   Application on keypoint detection

To further demonstrate the practical benefit of PDINet, we conduct an additional experiment on keypoint detection, a task that inherently involves projective distortions and thus provides a natural

Table 4: Memory usage and FLOPs of ResNet-18 and PDINet at different input resolutions.

| Input Size | Memory (MB) | | FLOPs | |
|---|---|---|---|---|
| | ResNet-18 | PDINet | ResNet-18 | PDINet |
| $32 \times 32$ | 0.53 | 0.93 | $3.72 \times 10^7$ | $3.65 \times 10^7$ |
| $64 \times 64$ | 2.10 | 3.73 | $1.49 \times 10^8$ | $1.46 \times 10^8$ |
| $128 \times 128$ | 8.38 | 14.91 | $5.94 \times 10^8$ | $5.84 \times 10^8$ |
| $256 \times 256$ | 33.50 | 59.63 | $2.38 \times 10^9$ | $2.33 \times 10^9$ |
| $512 \times 512$ | 134.00 | 238.50 | $9.51 \times 10^9$ | $9.34 \times 10^9$ |

Table 5: Runtime comparison on 1000 RGB images ($224 \times 224$) with batch size 100.

| Model | ResNet-18 | PDINet |
|---|---|---|
| Runtime (s) | $3.61_{\pm 0.35}$ | $6.84_{\pm 0.44}$ |

setting to evaluate the effectiveness of built-in projective equivariance. We integrate PDINet as the backbone into the REKD [Lee et al., 2022] framework for keypoint detection. To reduce confounding factors, we simplify the pipeline by removing the orientation estimation branch (which depends on steerable filters) and retain only the keypoint detection component. A three-layer PDINet is used and compared against a CNN baseline with the same architecture. The number of channels is adjusted to keep the parameter counts on the same order of magnitude, with PDINet using fewer parameters overall.

We evaluate both models on the viewpoint split of the HPatches dataset, which includes 59 scenes. Each scene contains a reference image and five target images captured from different viewpoints, resulting in projective distortions between image pairs. We follow the setup of Lee et al. [2022] for data construction and loss formulation, and train each model for 20 epochs using the AdamW optimizer with a cosine learning rate schedule (initial learning rate 0.01). A downsampling pyramid with scaling factor 1.2 is applied during training, and a symmetric pyramid with scaling factor $\sqrt{2}$ (plus an identity branch) is used at inference, with two levels of down-sampling and up-sampling.

We report the Repeatability metric, which measures the consistency of keypoint detection under viewpoint changes. A higher value indicates better robustness. As shown in Table 6, PDINet achieves higher repeatability than the baseline while using fewer parameters, suggesting that built-in projective equivariance enhances geometric consistency in keypoint detection. This supplementary experiment complements our main results and demonstrates the broader applicability of PDINet to real-world tasks.

Table 6: Keypoint detection results on the HPatches viewpoint split.

| Model | Repeatability $\uparrow$ | # Params |
|---|---|---|
| CNN | 42.1 | 8.3K |
| PDINet | 45.2 | 3.2K |

# F    Discussion

While PDINet consistently outperforms the baselines in our main experiments, some misclassifications still occur. We believe these errors are largely attributable to the nature of synthetically generated data. Synthetic projective transformations involve interpolation and padding, which can introduce aliasing, distortions, and unnatural edges, especially under strong shearing, scaling or non-orthogonal rotations. Such artifacts can interfere with the model's ability to maintain equivariance and thus degrade performance. In addition, since PDINet relies on differential invariants that depend on discrete approximations of derivatives, image resolution may also affect performance. For example, PDINet performs better on Proj-Imagenette than on Proj-STL-10, may partially attributed to the higher resolution enabling more accurate derivative estimation.

We expect that applying PDINet to real-world data at higher resolutions with naturally occurring projective distortions would help mitigate these artifacts and better demonstrate its full potential. In practical scenarios, such as multiview settings involving planar objects, transformations between different viewpoints are well modeled by projective mappings, making PDINet naturally suited for these cases. While exact equivariance may not strictly hold for non-planar or 3D objects, the projective inductive bias still contributes to improved robustness by approximately preserving geometric structure under near projective transformations. Exploring such extensions, including more general 3D settings, is a promising direction for future work.

Another avenue for exploration is to extend our method to other transformation groups. According to the general theory of differential invariants [Olver, 1993], such invariants exist for any regular Lie group action on a smooth manifold, for example, $SL(2, \mathbb{C})$ acting on the sphere, which corresponds to Möbius transformations. In principle, our framework can be adapted to these settings by deriving the appropriate differential invariants and designing corresponding equivariant architectures. While identifying explicit, concise, low-order, and numerically stable invariant forms for different groups (e.g., the Möbius group) is highly non-trivial, our experience suggests that recognizing relative invariants as atomic components can greatly facilitate the construction and simplification of a complete and practical invariant basis.

