# OpenReview forum: "Projective Equivariant Networks via Second-order Fundamental Differential Invariants"
_NeurIPS.cc/2025/Conference — NeurIPS 2025 spotlight_

### Official Review · Reviewer_rwYg · 2025-06-30

**Clarity:** 3
**Significance:** 3
**Originality:** 3
**Rating:** 5
**Confidence:** 5

**Summary:**

This paper presents a novel architecture equivariant under projective transformations (which are defined as the action of SL(3, R) on the plane via fractional linear transformations) based on differential features that are provably invariant. The paper focuses on the theoretical exposition of the recovery of the proposed family of differential invariants, though empirical evaluations are included. The appendix focuses on practical considerations regarding the implementation.

**Questions:**

See above.

**Ethical Concerns:**

["NO or VERY MINOR ethics concerns only"]

**Final Justification:**

This paper presents an original and scalable solution to handle projective equivariance.  Both the method's theoretical underpinnings and results are compelling, and the core module proposed by the authors is simple and can be used as a drop in layer in standard image networks.

I think this paper meets the bar for a clear accept and I will be happy to argue for this paper.

**Limitations:**

See above -- in particular, whether the generalization to other projective groups (such as SL(2, C) ) is not discussed.

This concern has now been satisfactorily addressed in the authors' rebuttal.

**Paper Formatting Concerns:**

None.

**Quality:**

3

**Strengths And Weaknesses:**

Overall, the authors method is relatively original, an extensive theoretical investigation is provided, and a simple neural module is proposed which serves as a building block and is empirically shown to be robust performance under the desired perturbations. On these criterion I feel this paper meets the bar for acceptance.

A few key weaknesses are as follows:

As stated, it is not clear the authors proposed method generalizes to other projective groups, the most obvious similar case being the action of SL(2, C) on the (Riemann) sphere. (This action has been previously studied in https://arxiv.org/pdf/2201.12212, which is closely related to the authors work -- appearing even to define a class of SL(2, C) moving frames to facilitate equivariance, see section 5.1 -- but is not discussed in the related work section and should be).

Most importantly, one of the authors main claims is that the proposed method is more efficient than previous approaches and is able to scale to higher resolution. This obviously appears to be true, and is demonstrated explicitly in experiments. However, the key practical benefit of this is that this should allow the model to actually be applied to real-world data. Unfortunately, such an experiment is lacking and in fact there is no evidence that projective equivariance is useful in practical problems other than being theoretically interesting. To be fair, such a motivation is not included in the work of MacLaughlan et al. nor any other paper I know of which seeks to take more challenging forms of equivariance. However, it begs the question of whether the authors work is necessary or useful from a practical perspective because the practical usefulness of projective equivariance is not justified in the literature.

Specifically, the efficiency of the method means it can actually be applied to real-world data, and showing it can be effective in those cases, and outperform existing methods, is extremely important and what this paper is missing. If the authors can show something convincing along these lines, I will increase my rating. Essentially, all the ingredients for success are here, but it's up to the authors to be more ambitious and creative in their applications. Simply saying that the network holds promise for vision applications in the conclusion is not enough and is potentially misleading.

---

> ### Author Rebuttal · Authors · 2025-07-31
>
> ### Weakness
>
> > **W1:** As stated, it is not clear the authors proposed method generalizes to other projective groups, the most obvious similar case being the action of SL(2, C) on the (Riemann) sphere. (This action has been previously studied in https://arxiv.org/pdf/2201.12212, which is closely related to the authors work -- appearing even to define a class of SL(2, C) moving frames to facilitate equivariance, see section 5.1 -- but is not discussed in the related work section and should be).
>
> **R1:**
> We thank the reviewer for highlighting this important work.
> We agree that the connection between our method and Möbius Convolutions (MC) [1] deserves explicit discussion.
> MC [1] achieve equivariance to spherical Möbius transformations and has demonstrated strong performance in geometry and spherical image processing tasks, particularly for triangular mesh datasets augmented with deformations.
> Both MC and our method aim to build equivariant models under certain projective group actions on specific geometric domains.
> However, the groups considered and the underlying geometric settings differ substantially.
> Our work considers SL(3, R) acting on the 2D projective plane: $x'=\frac{\alpha x+\beta y+\gamma}{\rho x+\sigma y+\tau},y'=\frac{\gamma x+\mu y+\nu}{\rho x+\sigma y+\tau}$, where $
> \begin{pmatrix}
> \alpha&\beta&\gamma\\\\
> \gamma&\mu&\nu\\\\
> \rho&\sigma&\tau
> \end{pmatrix}\in R^{3\times 3},x\in R,y\in R$,
> whereas MC considers SL(2, C): $z'=\frac{az+b}{cz+d}$ acting on the Riemann sphere through complex-valued Möbius transformations:
> $\begin{pmatrix}
> a&b\\\\
> c&d
> \end{pmatrix}\in C^{2\times 2},z\in C$.
> Thus, while both fall under the umbrella of projective geometry, the symmetry groups and domains of action, planar vs. spherical, are distinct, as are the types of input data they address (planar images vs. spherical signals).
> Additionally, the "frame operator" in MC is used to define how filters transform at each point on the sphere to satisfy Möbius equivariance, whereas the "moving frame" concept we introduce is used to compute differential invariants of the projective group.
>
> Regarding generalizability, Theorem 2.17 in [2] guarantees the existence of differential invariants for any regular Lie group action on a smooth manifold with finite-dimensional orbits.
> This applies not only to SL(3, R) acting on functions over the 2D plane, but also to other projective groups such as SL(2, C) acting on functions over the sphere.
> Therefore, in principle, our method is extendable to these settings.
> Actually, the spherical setting also aligns with prior works such as [3], which explore equivariant partial differential operators (PDOs) on spherical domains.
> Their results indicate that both derivative estimation and invariant construction are practically feasible in such contexts.
> However, while the existence of differential invariants is well established, identifying explicit, concise, low-order, and numerically stable invariant forms for different groups (e.g., Möbius) is highly non-trivial.
> In addition, how to design architectures that effectively leverage such invariants across different domains remains to be explored.
>
> We will include the above discussion in the revised related work section.
> Thank you again for bringing up this important point.
>
>
>
>
>
>
> > **W2:** Most importantly, one of the authors main claims is that the proposed method is more efficient than previous approaches and is able to scale to higher resolution. This obviously appears to be true, and is demonstrated explicitly in experiments. However, the key practical benefit of this is that this should allow the model to actually be applied to real-world data. Unfortunately, such an experiment is lacking and in fact there is no evidence that projective equivariance is useful in practical problems other than being theoretically interesting. To be fair, such a motivation is not included in the work of MacLaughlan et al. nor any other paper I know of which seeks to take more challenging forms of equivariance. However, it begs the question of whether the authors work is necessary or useful from a practical perspective because the practical usefulness of projective equivariance is not justified in the literature.
>
> > Specifically, the efficiency of the method means it can actually be applied to real-world data, and showing it can be effective in those cases, and outperform existing methods, is extremely important and what this paper is missing. If the authors can show something convincing along these lines, I will increase my rating. Essentially, all the ingredients for success are here, but it's up to the authors to be more ambitious and creative in their applications. Simply saying that the network holds promise for vision applications in the conclusion is not enough and is potentially misleading.
>
> **R2:**
> Thank you for the detailed and thoughtful feedback.
>
> Projective transformations are fundamental in computer vision, as they describe common phenomena such as perspective distortion and viewpoint variation.
> Incorporating projective equivariance into network design offers a principled way to embed such geometric priors, enabling models to generalize under these transformations.
> We believe this makes projective equivariance a meaningful and worthwhile direction to explore, both theoretically and practically.
>
> We fully agree that demonstrating real-world utility is essential for establishing the practical value of this approach.
> However, progress in this challenging area has been gradual.
> The first projective equivariant model, homConv [4], is limited to shallow networks and low-resolution grayscale inputs (e.g., MNIST) due to exponential memory requirements.
> Our work advances the field by enabling deeper networks and supporting high-resolution RGB images, allowing evaluation on realistic datasets including STL-10 and Imagenette.
> While further applications may require task-specific adaptations, we believe our contributions significantly improve scalability and bring projective equivariant networks closer to real-world deployment.
>
> Motivated by your suggestion, we go a step further and conduct a preliminary experiment demonstrating the practical benefit of PDINet in the context of a real-world task: keypoint matching.
> This task inherently involves projective distortions and is a natural fit evaluate the benefits of built-in projective equivariance.
> Specifically, we integrate PDINet as the backbone into the REKD [5] framework for keypoint detection.
> To reduce confounding factors, we simplify the pipeline by removing the orientation estimation branch (which depends on steerable filters) and retain only the keypoint detection component.
> We use a three-layer PDINet and compare it against a CNN baseline of the same architecture.
> The number of channels is adjusted to keep the parameter counts on the same order of magnitude, with PDINet using fewer parameters overall.
>
> We evaluate both models on the viewpoint split of the HPatches dataset, which includes 59 scenes.
> Each scene contains a reference image and five target images captured from different viewpoints, resulting in projective distortions between image pairs.
> Both models are trained for 20 epochs using the AdamW optimizer with a cosine learning rate schedule with initial learning rate 1e-2.
> Following the setup in [5], we use the same training data construction and loss function, and apply a downsampling pyramid with scaling factor $1.2$ during training.
> At inference, we employ a symmetric downsampling and upsampling pyramid with scaling factor $\sqrt{2}$, alongside an unscaled identity branch.
> The number of down-sampling and up-sampling are both set to two.
>
> We report the Repeatability metric, which measures the consistency of keypoint detection under viewpoint changes.
> A higher value indicates better robustness.
> As shown in Table 1, PDINet achieves higher repeatability than the baseline while using fewer parameters, suggesting that built-in projective equivariance enhances geometric consistency in keypoint detection.
>
> We sincerely appreciate your suggestion, which help us expand our experimental scope beyond classification.
> We believe this initial result demonstrates the feasibility and promise of applying PDINet to real-world vision tasks, and we look forward to exploring additional applications in future work.
>
> **Table 1**: Keypoint detection results on the HPatches viewpoint split.
> | Model    | Repeatability ↑ | # Params |
> | -------- | --------------- | ------------ |
> | CNN      | 42.1            | 8.3 K         |
> | PDINet   | 45.2            | 3.2 K         |
>
>
>
> ### Limitations
>
> > **L1:** in particular, whether the generalization to other projective groups (such as SL(2, C) ) is not discussed.
>
> **R1:**
> Please refer to our response to W1.
>
>
> [1] Mitchel et al. Möbius convolutions for spherical CNNs (2022)
>
> [2] Peter J. Olver. Applications of Lie Groups to Differential Equations (1993)
>
> [3] Shen et al. PDO-eS2CNNs: Partial differential operator based equivariant spherical CNNs (2021)
>
> [4] MacDonald et al. Enabling equivariance for arbitrary Lie groups (2022)
>
> [5] Lee et al. Self-Supervised Equivariant Learning for Oriented Keypoint Detection (2022)

---

> ### Comment · Reviewer_rwYg · 2025-08-04
>
> Thanks for your detailed, thorough and refreshing rebuttal. I especially appreciate the experiments on real-world data, and will increase my score to a clear accept.
>
> That said, I have a few remaining questions/comments I hope the authors can address.
>
> 1). In regards to generalization to other projective groups, such as SL(2, C), etc., your response indicates that to generalize the proposed method several considerations specific to the new group and homogeneous space (e.g. the specific invariants) are necessary. From my understanding, it appears that your proposed layer is readily generalizable, since it basically consists of taking a derivative, combining the results in a specific way, and passing it through an MLP. This should generalize to any arbitrary domain (as long as you can numerically well-define differentiation) and it appears the limiting factor is finding the invariants themselves.
>
> Using SL(2, C) acting on the (Riemman) sphere as an example, could you please sketch out the necessary steps to generalize your proposed approach to this domain?
>
> I ask this only because most existing equivariant networks are handcrafted, and have to be re-engineered to handle new classes of groups.
>
> 2). I also agree with other reviewers that a comparison with a data-augmented baseline would be nice for the sake of completeness, though I disagree that it is necessary for acceptance. Rather, if the authors can find the time between now and the camera ready deadline to perform such an experiment and report the results, that would be ideal.
>
> In any case, I congratulate the authors on a fine paper which tackles a interesting and challenging problem in an original way!

---

> > ### Author Response · Authors · 2025-08-05
> > **Response to Reviewer rwYg**
> >
> > We sincerely appreciate your recognition of our work and your insightful question.
> >
> > A key strength of our approach is its systematic and generalizable pipeline, which can, in principle, be extended to a broad range of transformation groups.
> > As you pointed out, once suitable differential invariants are derived and numerical derivative estimation is feasible, the network design in our work can be readily adapted to new domains, including the Riemann sphere under the action of SL(2, C).
> > Thus, the essential step in such a generalization lies in deriving the appropriate differential invariants.
> > To achieve this, one could follow a similar procedure as in our work:
> >
> > 1. Compute the transformation behavior of derivatives under the SL(2, C) action.
> > This typically involves characterizing how first- and at least second-order derivatives of functions on the sphere transform under Möbius transformations.
> >
> > 2. Select a suitable cross section for the group action and apply the moving frame method to derive a complete set of fundamental differential invariants.
> > We recommend using a multi-channel and low-order cross section, which may allow for a more manageable set of invariants.
> >
> > 3. Simplify and reorganize the resulting invariants to seek a concise and low-degree generating set.
> > In practice, identifying relative invariants as atomic components often facilitates the construction of a complete invariant basis.
> >
> > We hope this provides a useful sketch for extending the framework to broader settings, and we appreciate your interest in its extensibility.
> >
> >
> > We also agree that reporting the ResNet baseline with data augmentation (DA) contributes to a more complete evaluation.
> > In response to the reviewers’ suggestion, we conduct additional experiments, and present the performance of ResNet-18 trained with projective data augmentations in Tables 2 and 3, reported as the mean and standard deviation over five runs.
> > While data augmentation substantially improves ResNet’s performance, our model still slightly outperforms it on both datasets.
> >
> > Thank you again for your thoughtful feedback and support.
> >
> > **Table 2:** Test accuracy (\%) on Proj-STL-10.
> > | Model                      | Accuracy  | # Params  |
> > |----------------------------|----------------|------------------|
> > | ResNet-18                  | $39.73 \pm 0.33$ | $11.18$ M  |
> > | ResNet-18 with DA          | $48.73 \pm 0.53$ | $11.18$ M  |
> > | homConv                    | $21.26$          | $376$ K  |
> > | PDINet (ours)              | $51.12 \pm 0.47$ | $8.90$ M  |
> >
> > **Table 3:** Test accuracy (\%) on Proj-Imagenette.
> > | Model                      | Accuracy  | # Params  |
> > |----------------------------|----------------|------------------|
> > | ResNet-18                  | $65.64 \pm 0.39$ | $11.18$ M  |
> > | ResNet-18 with DA          | $70.77 \pm 0.76$ | $11.18$ M  |
> > | homConv                    | $26.57$          | $376$ K  |
> > | PDINet (ours)              | $71.30 \pm 0.45$ | $8.90$ M  |

---

> > > ### Comment · Reviewer_rwYg · 2025-08-05
> > >
> > > Thank you for the detailed explanation and for including the additional baseline experiments.
> > >
> > > I am very happy with this paper and will champion its acceptance!

---

> > > > ### Author Response · Authors · 2025-08-06
> > > > **Response to Reviewer rwYg**
> > > >
> > > > We truly value your recognition and feedback, and we will incorporate these additions into the revised version of the paper.

---

### Official Review · Reviewer_VbGP · 2025-07-02

**Clarity:** 3
**Significance:** 3
**Originality:** 3
**Rating:** 4
**Confidence:** 3

**Summary:**

This paper demonstrates a novel method to design a projective equivariant neural network. It is based on finding the projective differential invariants explicitly and use these features to perform learning. The experimental result demonstrates its effectiveness and improvements.

**Questions:**

1. What is the computational complexity/cost of these operations?
2. Would the authors extend experiment/baseline study? How to verify the exact equivariance experimentally?
3. What type of changes would happen when we act a random projective group action to an image? Would the authors be able to show a few example visualizations in the main paper (or the appendix)?
4. Could the authors be sure to include more equivariant network literature?

**Ethical Concerns:**

["NO or VERY MINOR ethics concerns only"]

**Final Justification:**

With added experiments the reviewer is clear that this is a valid contribution in this community. As a research who had a few works on this domain, my comment can be a bit conservative, but the reviewer believes that this 'signature-like' method is valid and can be useful.

**Limitations:**

Yes.

**Paper Formatting Concerns:**

No.

**Quality:**

3

**Strengths And Weaknesses:**

Strengths:

1. Projective equivariance is an important type of equivariance, and there have not been many works addressing this. Having a projective equivariant design can be important for general vision problems. The significance and originality of this paper are good.
2. This paper is theoretically strong, and the reviewer believes the derivation of differential invariants is correct.
3. The experimental results demonstrate the initial performance and improvement of this method.

Weaknesses:

1. This paper is not formulated in its best. The derivation is important but could be moved to the appendix. The reviewer suggests that the authors move Figure 1 earlier in the paper.
2. The paper demonstrates a promising strategy to enable projective equivariance, but the claim is not fully verified. The authors should find a way to visualize and verify that the map is equivariant. It is hard to fully understand what causes the improvement.
3. The OOM issue might be resolved either by using a larger device or by reducing the sample size (if the reviewer understands the OOM error correctly).
4. The experimental study and baseline comparison are relatively weak.

---

> ### Author Rebuttal · Authors · 2025-07-31
>
> ### Weakness
>
> > **W1:** This paper is not formulated in its best. The derivation is important but could be moved to the appendix. The reviewer suggests that the authors move Figure 1 earlier in the paper.
>
> **R1:**
> Thank you for the suggestion.
> In the next version, we will reduce the detailed derivations in the main text and move Figure 1 earlier in the paper.
>
>
>
> > **W2:** The paper demonstrates a promising strategy to enable projective equivariance, but the claim is not fully verified. The authors should find a way to visualize and verify that the map is equivariant. It is hard to fully understand what causes the improvement.
>
> **R2:**
> Theoretical verification: The projective equivariance of our operators is rigorously guaranteed by the fundamental properties of differential invariants, as shown by the equation:
> \[\hat{I}(g \cdot u)(x) = I(x, g \cdot u) = I(g^{-1} \cdot x, u) = \hat{I}(u)(g^{-1} \cdot x) = (g \cdot \hat{I}(u))(x)\].
>
> Experimental validation: To quantitatively verify equivariance, we follow the approach in [1] to test the equivariance error, which is defined as:
> \[ \text{Error} = \frac{\| g \cdot \psi(u) - \psi(g \cdot u) \|^2}{\| g \cdot \psi(u) \|^2} \],
> where \( g \) is a random projective transformation, \( u \) is the input image, and \( \psi \) is the equivariant layer.
> We computed this error on Imagenette test images resized to multiple resolutions (Table 1).
> Results demonstrate a monotonic decrease in equivariance error with increasing image size, which is primarily due to the more accurate estimation of derivatives at higher resolutions.
>
> Visualization: We acknowledge the value of qualitative demonstrations.
> In the next version, we will add visualizations comparing features $g \cdot \psi(u) $and $\psi(g \cdot u)$ to illustrate equivariance more intuitively.
>
>
> **Table 1**: Equivariance error.
> | image size         | $16 \times 16$   | $32 \times 32$   | $64 \times 64$   | $128 \times 128$ | $256 \times 256$ |
> |--------------------|------------------|----------------|------------------|------------------|------------------|
> | equivariance error (\%) | 0.34             | 0.14           | 0.08             | 0.04             | 0.02             |
>
>
>
>
>
>
> > **W3:** The OOM issue might be resolved either by using a larger device or by reducing the sample size (if the reviewer understands the OOM error correctly).
>
> **R3:**
> To enable a direct numerical comparison with homConv, we follow the original network structure used in [1] and reduce the number of samples to avoid out-of-memory (OOM) issues.
> The experimental results are shown in Tables 2 and 3.
> Due to its restriction on network depth, homConv struggles to perform well on image datasets with higher resolutions.
>
> **Table 2:** Test accuracy (\%) on Proj-STL-10 after training on STL-10.
> | Model                      | Accuracy  | # Params  |
> |----------------------------|----------------|------------------|
> | ResNet-18                  | $39.73 \pm 0.33$ | $11.18$ M  |
> | homConv                    | $21.26$          | $376$ K  |
> | PDINet (ours)              | $51.12 \pm 0.47$ | $8.90$ M  |
>
> **Table 3:** Test accuracy (\%) on Proj-Imagenette after training on Imagenette.
> | Model                      | Accuracy  | # Params  |
> |----------------------------|----------------|------------------|
> | ResNet-18                  | $65.64 \pm 0.39$ | $11.18$ M  |
> | homConv                    | $26.57$          | $376$ K  |
> | PDINet (ours)              | $71.30 \pm 0.45$ | $8.90$ M  |
>
>
>
> > **W4:** The experimental study and baseline comparison are relatively weak.
>
> **R4:**
> Thank you for your feedback.
> Projective equivariance remains a challenging problem, and to our knowledge, homConv [1] is the first and, so far, the only existing equivariant model designed for projective symmetry on the 2D plane.
> Therefore, we adopt it as our primary point of comparison.
> Following your suggestion, we have added a direct numerical comparison with homConv.
> Our experimental study is consistent with that of homConv and follows the out-of-distribution (OOD) evaluation protocol, which tests a model’s ability to generalize to projectively transformed inputs that are not seen during training.
> Under this evaluation setting, PDINet consistently outperforms a ResNet with the same architecture and scales more effectively than homConv, highlighting its advantages in both robustness and scalability.
>
> In addition, responding to suggestions from other reviewers, we have further extended our experimental evaluation to demonstrate the practical benefit of PDINet in the context of keypoint matching.
> This task inherently involves projective distortions and is a natural fit evaluate the benefits of built-in projective equivariance.
> Specifically, we integrate PDINet as the backbone into the REKD [2] framework for keypoint detection.
> To reduce confounding factors, we simplify the pipeline by removing the orientation estimation branch (which depends on steerable filters) and retain only the keypoint detection component.
> We use a three-layer PDINet and compare it against a CNN baseline of the same architecture.
> The number of channels is adjusted to keep the parameter counts on the same order of magnitude, with PDINet using fewer parameters overall.
>
> We evaluate both models on the viewpoint split of the HPatches dataset, which includes 59 scenes.
> Each scene contains a reference image and five target images captured from different viewpoints, resulting in projective distortions between image pairs.
> Both models are trained for 20 epochs using the AdamW optimizer with a cosine learning rate schedule with initial learning rate 1e-2.
> Following the setup in [2], we use the same training data construction and loss function, and apply a downsampling pyramid with scaling factor $1.2$ during training.
> At inference, we employ a symmetric downsampling and upsampling pyramid with scaling factor $\sqrt{2}$, alongside an unscaled identity branch.
> The number of down-sampling and up-sampling are both set to two.
>
> We report the Repeatability metric, which measures the consistency of keypoint detection under viewpoint changes.
> A higher value indicates better robustness.
> As shown in Table 4, PDINet achieves higher repeatability than the baseline while using fewer parameters, suggesting that built-in projective equivariance enhances geometric consistency in keypoint detection.
> This experiment complements our main results and demonstrates the broader applicability of PDINet to real-world tasks.
>
> **Table 4**: Keypoint detection results on the HPatches viewpoint split.
> | Model    | Repeatability ↑ | # Params |
> | -------- | --------------- | ------------ |
> | CNN      | 42.1            | 8.3 K         |
> | PDINet   | 45.2            | 3.2 K         |
>
>
>
>
> ### Questions
>
>
>
>
> > **Q1:** What is the computational complexity/cost of these operations?
>
> **A1:**
> Our projective equivariant layer exhibits linear growth in both time and space complexity with respect to the input size.
> Specifically, the computation include estimating derivatives, computing differential invariants, and combining differential invariants, each of which has linear complexity.
> We numerically measure the computational cost of a single projective equivariant layer and report the results in Table 5.
> We use \emph{torchstat} to calculate the memory usage and the FLOPs required for model inference.
> As shown in Table 5, both the memory usage and the FLOPs scale linearly with the input size.
>
> **Table 5**: The FLOPs and memory occupation of a projective equivariant layer.
> | Input Size     | Memory (MB) | FLOPs          |
> |----------------|-------------|----------------|
> |$32 \times 32$  | $0.02$      |$5.53\times10^4$|
> |$64 \times 64$  | $0.09$      |$2.21\times10^5$|
> |$128\times128$  | $0.38$      |$8.84\times10^5$|
> |$256\times256$  | $1.50$      |$3.54\times10^6$|
> |$512\times512$  | $6.00$      |$1.42\times10^7$|
>
>
>
>
>
>
> > **Q2:** Would the authors extend experiment/baseline study? How to verify the exact equivariance experimentally?
>
> **A2:**
> Please refer to our responses to W2, W3 and W4.
>
>
>
>
>
>
> > **Q3:** What type of changes would happen when we act a random projective group action to an image? Would the authors be able to show a few example visualizations in the main paper (or the appendix)?
>
> **A3:**
> A projective transformation preserves straight lines but introduces perspective distortion, such as changes in relative scale, loss of parallelism, and nonlinear deformation of shapes.
> For example, a square may appear as a trapezoid or an arbitrary quadrilateral under a projective transformation.
> These transformations naturally arise in multi-view settings, where a planar object is projected onto different image planes due to changes in viewpoint or camera pose.
> Appendix A includes a conceptual illustration of a projective transformation, showing how such distortions occur when projecting a planar object onto another plane.
> For concrete examples, Figure 1 in [1] shows how projective transformations affect images in practice.
> We will also include additional visualization examples in the next revision to help readers better understand the visual effects of projective transformations on natural images.
>
>
>
>
>
> > **Q4:** Could the authors be sure to include more equivariant network literature?
>
> **A4:**
> Thank you for the suggestion.
> We include a section of related works on equivariant networks in Appendix B.2, covering both discrete and continuous group equivariance approaches.
> In addition, in response to Reviewer rwYg’s helpful comment, we will expand the related work section to include the study in [3], which explores equivariant networks under the action of SL(2, C) on the Riemann sphere —- a different type of projective group action.
>
>
> [1] MacDonald et al. Enabling equivariance for arbitrary Lie groups (2022)
>
> [2] Lee et al. Self-Supervised Equivariant Learning for Oriented Keypoint Detection (2022)
>
> [3] Mitchel et al. Möbius convolutions for spherical CNNs (2022)

---

> > ### Comment · Reviewer_VbGP · 2025-08-08
> >
> > Incredible response and it resolved most of my concerns. Please be sure to put the updated experimental study in its final form.
> >
> > That said, the experiments could be more extensive - for the current work or future investigation.
> >
> > Good luck with the paper and the reviewer will update the score.

---

> > > ### Author Response · Authors · 2025-08-09
> > >
> > > We greatly appreciate your thoughtful comments and feedback, and will incorporate the updated experimental study in the final version.
> > > We will further investigate the experiments to explore the potential of our method.

---

> ### Author Response · Authors · 2025-08-06
>
> Dear Reviewer VbGP,
>
> We sincerely thank you for your thoughtful comments and suggestions.
>
> We have prepared a detailed response to address your questions.
> Additionally, we have included ResNet-18 trained with data augmentation as a baseline for comparison (please refer to Tables 2 and 3 in our response to Reviewer rwYg).
> If there are any aspects you would like us to clarify, we look forward to further discussion.
>
> We greatly appreciate your time and feedback.
>
> Best regards,
>
> Authors

---

> ### Comment · Area_Chair_Jh9D · 2025-08-07
>
> Dear reviewer VbGP, please discuss with the authors. Did they address your concerns, or do you have other questions? To note, in this year's NeurIPS, discussions are required before the mandatory acknowledgement.

---

> > ### Comment · Reviewer_VbGP · 2025-08-08
> >
> > Found myself overloaded with all the papers in my review batch... Sorry for the late. They had a great response and I recommend acceptance.
> >
> > Best,
> >
> > VbGP

---

### Official Review · Reviewer_6v2q · 2025-07-03

**Clarity:** 3
**Significance:** 3
**Originality:** 4
**Rating:** 5
**Confidence:** 3

**Summary:**

This paper introduces a deep neural network equivariant to projective transformations. The core contribution is the theoretical derivation of a complete and concise set of second-order fundamental differential invariants for the projective group. These invariants are derived using moving frames with a novel cross-section for multi-dimensional functions (e.g., multi-channel feature maps), which allows the use of second-order derivatives, unlike previous approaches for scalar functions that required more complex higher-order derivatives. The resulting neural network is sampling-free and thus does not have the scalability problem as previous work.

**Questions:**

Please address the mentioned weaknesses.

Besides, how are first and second-order partial derivatives computed in practice within the network? Does the pixel discretization affect the accuracy of the computed invariants and the overall performance of the network?

**Ethical Concerns:**

["NO or VERY MINOR ethics concerns only"]

**Final Justification:**

The paper's theoretical and technical contributions are outstanding. After providing the extra information and experimental results, I recommend accepting this paper.

**Limitations:**

yes

**Quality:**

3

**Strengths And Weaknesses:**

Strengths:

The paper could be the first work that tackles the problem of developing neural networks with projective equivariance, which is common in camera geometry, in a scalable way. The theoretical framework based on differential invariants is novel and solid. The framework leads to a scalable network implementation, which is impressive.

The paper is well-structured and well-written. The paper requires a substantial mathematical background to follow, but the authors did a good job explaining the background and the derivations.

Weaknesses:

Limited Experimental Comparisons: The primary baseline is a standard ResNet-18, which is not designed for geometric robustness. While the OOM issue with homConv is a valid point about scalability, the lack of any direct numerical comparison to it is a weakness. It would strengthen the paper to include a comparison on a shallower network where homConv could run. Furthermore, a comparison to a strong data augmentation baseline (i.e., ResNet-18 trained on projectively augmented data) is missing. Such a comparison helps disentangle the benefits of the equivariant architecture from the benefits of simply exposing the model to the test distribution's transformations during training.

Computational Cost Analysis: The paper argues that its invariants are more concise and computationally efficient than alternatives. However, it lacks a practical analysis of the computational overhead. How does it compare with the original ResNet-18 in terms of training/inference time, memory consumption, and FLOPs?

---

> ### Author Rebuttal · Authors · 2025-07-31
>
> ### Weakness
>
>
> > **W1:** Limited Experimental Comparisons: The primary baseline is a standard ResNet-18, which is not designed for geometric robustness. While the OOM issue with homConv is a valid point about scalability, the lack of any direct numerical comparison to it is a weakness. It would strengthen the paper to include a comparison on a shallower network where homConv could run. Furthermore, a comparison to a strong data augmentation baseline (i.e., ResNet-18 trained on projectively augmented data) is missing. Such a comparison helps disentangle the benefits of the equivariant architecture from the benefits of simply exposing the model to the test distribution's transformations during training.
>
> **R1:**
> Thank you for the valuable suggestions.
>
> To enable a direct numerical comparison with homConv, we follow the shallow network structure used in [1] and reduce the number of samples used to approximate convolution and pooling, thus avoiding out-of-memory (OOM) issues.
> The experimental results are shown in Tables 1 and 2.
> Due to its restriction on network depth, homConv struggles to perform well on image datasets with higher resolutions.
>
> We also agree that comparisons with data augmentation baselines are common in other equivariant network studies.
> However, our work adopts the out-of-distribution (OOD) evaluation setup used by homConv [1], which is the most relevant prior work targeting projective equivariance and serves as our primary point of comparison.
> This evaluation setup is designed to test whether a model can generalize to projectively transformed inputs without access to the specific distribution of test-time transformations.
> While our model incorporates prior knowledge of the symmetry group (i.e., the projective group), it does not rely on assumptions about the specific range or distribution of transformation parameters.
> In contrast, data augmentation approaches require explicitly specifying and sampling from these parameter spaces, which are often unknown or difficult to specify in practice.
> We believe this evaluation setup remains consistent with prior work [1] and is well-suited for assessing the intended capabilities of projective equivariant networks.
>
>
> **Table 1:** Test accuracy (\%) on Proj-STL-10 after training on STL-10.
> | Model                      | Accuracy  | # Params  |
> |----------------------------|----------------|------------------|
> | ResNet-18                  | $39.73 \pm 0.33$ | $11.18$ M  |
> | homConv                    | $21.26$          | $376$ K  |
> | PDINet (ours)              | $51.12 \pm 0.47$ | $8.90$ M  |
>
> **Table 2:** Test accuracy (\%) on Proj-Imagenette after training on Imagenette.
> | Model                      | Accuracy  | # Params  |
> |----------------------------|----------------|------------------|
> | ResNet-18                  | $65.64 \pm 0.39$ | $11.18$ M  |
> | homConv                    | $26.57$          | $376$ K  |
> | PDINet (ours)              | $71.30 \pm 0.45$ | $8.90$ M  |
>
>
>
>
>
>
> > **W2:** Computational Cost Analysis: The paper argues that its invariants are more concise and computationally efficient than alternatives. However, it lacks a practical analysis of the computational overhead. How does it compare with the original ResNet-18 in terms of training/inference time, memory consumption, and FLOPs?
>
> **R2:**
> The conciseness of our invariants is relative to the invariants derived using the method in [2], which involves third-order derivatives.
> For example, the projective invariants for scalar functions in [2] contain dozens of terms with degrees as high as 12.
> Extending such a method to the multi-channel setting would result in significantly more complex expressions with hundreds of terms.
> In contrast, our derived invariants only involve up to second-order derivatives, with polynomial degrees no greater than 3 and fewer than 8 terms.
> This conciseness facilitates both theoretical analysis and practical applications.
>
> Regarding computational overhead, we thank the reviewer for the suggestion, as it highlights the scalability of our model to higher-resolution inputs.
> Our projective equivariant layer exhibits linear growth in both time and space complexity with respect to the input size.
> Specifically, the computation includes estimating derivatives, computing differential invariants, and combining differential invariants, each of which has linear complexity.
> To quantify the overhead, we compare PDINet and ResNet-18 in terms of memory usage and FLOPs using \emph{torchstat}, and report the results in Table 3.
> As shown, both models exhibit linear scaling with respect to input resolution.
> PDINet has slightly higher memory usage than ResNet-18, but requires fewer FLOPs.
> In practice, PDINet's inference speed is slower than ResNet-18, despite the lower FLOPs.
> This is primarily because convolutional layers in ResNet-18 benefit from highly optimized low-level libraries, whereas our custom layers currently do not.
> We will try to develop a CUDA-accelerated implementation in future work to further improve training and inference efficiency.
>
> **Table 3**: The FLOPs and memory occupation of ResNet-18 and PDINet.
> | Input Size      | ResNet-18               | PDINet                |
> |-----------------|------------------------|-----------------------|
> |                | **Memory (MB)**        | **Memory (MB)**       |
> | $32\times 32$   | 0.53                   | 0.93                  |
> | $64\times 64$   | 2.10                   | 3.73                  |
> | $128\times 128$ | 8.38                   | 14.91                 |
> | $256\times 256$ | 33.50                  | 59.63                 |
> | $512\times 512$ | 134.00                 | 238.50                |
> |                | **FLOPs**              | **FLOPs**             |
> | $32\times 32$   | $3.72\times 10^7$      | $3.65\times 10^7$     |
> | $64\times 64$   | $1.49\times 10^8$      | $1.46\times 10^8$     |
> | $128\times 128$ | $5.94\times 10^8$      | $5.84\times 10^8$     |
> | $256\times 256$ | $2.38\times 10^9$      | $2.33\times 10^9$     |
> | $512\times 512$ | $9.51\times 10^9$      | $9.34\times 10^9$     |
>
>
>
>
> ### Questions
>
> > **Q1:** Besides, how are first and second-order partial derivatives computed in practice within the network? Does the pixel discretization affect the accuracy of the computed invariants and the overall performance of the network?
>
> **A1:**
> As described in Appendix D, we estimate partial derivatives using Gaussian derivatives.
> For example, the partial derivative with respect to \( x \) is computed via convolution as:
> \[ \frac{\partial f}{\partial x} \approx f \ast \frac{\partial G_\sigma}{\partial x} \],
> where \( G_\sigma \) is a Gaussian kernel with zero mean and standard deviation \( \sigma \).
>
> Pixel discretization does affect the accuracy of derivative computations,
> which in turn influence the computation of differential invariants and, consequently, the overall equivariance of the network.
> To quantify this effect, we verify equivariance by following the approach in [1] to test the equivariance error, which is defined as:
> \[ \text{Error} = \frac{\| g \cdot \psi(u) - \psi(g \cdot u) \|^2}{\| g \cdot \psi(u) \|^2} \],
> where \( g \) is a random projective transformation, \( u \) is the input image, and \( \psi \) is the equivariant layer.
> We computed this error on Imagenette test images resized to multiple resolutions, as shown in Table 4.
> The results indicate that the equivariance error remains small across resolutions and consistently decreases as image resolution increases, which is expected due to more accurate derivative approximation at higher resolutions.
>
> **Table 4**: Equivariance error.
> | image size         | $16 \times 16$   | $32 \times 32$   | $64 \times 64$   | $128 \times 128$ | $256 \times 256$ |
> |--------------------|------------------|----------------|------------------|------------------|------------------|
> | equivariance error (\%) | 0.34             | 0.14           | 0.08             | 0.04             | 0.02             |
>
>
>
>
> [1] MacDonald et al. Enabling equivariance for arbitrary Lie groups (2022)
>
> [2] Peter J. Olver. Projective invariants of images (2023)

---

> > ### Comment · Reviewer_6v2q · 2025-08-03
> > **Still missing: comparison to ResNet with data augmentation and time efficiency**
> >
> > The added information is valuable, but two important results are still missing.
> >
> > First, the ResNet-18 result with data augmentation is still not reported. Previous work not reporting this result is not a valid reason for following-up works also skipping that result. There is no doubt that comparing equivariant networks to non-equivariant methods with data augmentation is essential for assessing the value, potential, and limitations of equivariant methods, and also a standard practice. You can include ResNet's results both with and without augmentation, which will still keep your results comparable to homConv while being more holistic. My personal experience is that a generic ResNet can handle projective transformations reasonably well when trained with augmentation. Whether that's the case or not, honestly reporting that is important.
> >
> > Second, the time efficiency comparison is not reported, which is also an important aspect for general readers to understand the practicality of this approach.
> >
> > I won't be disappointed if either of the two results is not in favor of the proposed method, as the core value of this paper is in the new method derived from the theoretical framework of differential invariants, which is novel and achieves guaranteed projective equivariance with unprecedented accuracy and efficiency. But I will only consider accepting the paper if these two results are reported and included in the final paper.

---

> > > ### Author Response · Authors · 2025-08-05
> > > **Response to Reviewer 6v2q**
> > >
> > > We sincerely thank the reviewer for the constructive follow-up.
> > > We agree that reporting both the ResNet baseline with data augmentation (DA) and an explicit runtime comparison contributes to a more complete evaluation.
> > > Following your suggestions, we conduct additional experiments and now report both results.
> > >
> > > The performance of ResNet-18 trained with projective data augmentations is presented in Tables 5 and 6.
> > > Each result is reported as the mean and standard deviation over five runs.
> > > While data augmentation substantially improves ResNet’s performance, our model still slightly outperforms it on both datasets.
> > >
> > > To provide an explicit runtime comparison, we evaluate PDINet and ResNet-18 on the same hardware by measuring the total time for processing 1000 RGB images ($224 \times 224$) with batch size 100.
> > > Each experiment is repeated five times, and we report the mean and standard deviation in Table 7.
> > > Despite its lower theoretical FLOPs (Table 3), PDINet runs slower in practice.
> > > This is primarily due to the highly optimized low-level implementations of ResNet-18.
> > > We believe the gap can be narrowed through further engineering and implementation optimization.
> > >
> > > We thank the reviewer again for these valuable suggestions and will include all additional results in the final version of the paper.
> > >
> > > **Table 5:** Test accuracy (\%) on Proj-STL-10.
> > > | Model                      | Accuracy  | # Params  |
> > > |----------------------------|----------------|------------------|
> > > | ResNet-18                  | $39.73 \pm 0.33$ | $11.18$ M  |
> > > | ResNet-18 with DA          | $48.73 \pm 0.53$ | $11.18$ M  |
> > > | homConv                    | $21.26$          | $376$ K  |
> > > | PDINet (ours)              | $51.12 \pm 0.47$ | $8.90$ M  |
> > >
> > > **Table 6:** Test accuracy (\%) on Proj-Imagenette.
> > > | Model                      | Accuracy  | # Params  |
> > > |----------------------------|----------------|------------------|
> > > | ResNet-18                  | $65.64 \pm 0.39$ | $11.18$ M  |
> > > | ResNet-18 with DA          | $70.77 \pm 0.76$ | $11.18$ M  |
> > > | homConv                    | $26.57$          | $376$ K  |
> > > | PDINet (ours)              | $71.30 \pm 0.45$ | $8.90$ M  |
> > >
> > >
> > > **Table 7:** Runtime comparison on 1000 RGB images ($224 \times 224$) with batch size 100.
> > > | Model         | Runtime (s)       |
> > > | ------------- | ----------------- |
> > > | ResNet-18     | $3.61 \pm 0.35$ |
> > > | PDINet (ours) | $6.84 \pm 0.44$ |

---

> > > > ### Comment · Reviewer_6v2q · 2025-08-08
> > > >
> > > > Thank you for providing these results. My concerns are resolved. Please include the new results in the final version of the paper. I think this paper made a significant contribution to the field of equivariant deep learning.

---

> > > > > ### Author Response · Authors · 2025-08-08
> > > > >
> > > > > We sincerely thank you for your constructive feedback and recognition of our work.
> > > > > We will include the new results in the final version of the paper.

---

### Official Review · Reviewer_gqa7 · 2025-07-03

**Clarity:** 3
**Significance:** 3
**Originality:** 3
**Rating:** 5
**Confidence:** 3

**Summary:**

This paper intrdocues PDINet, a network architecture that is equivariant to homographies. The key idea is to construct differential invariants of the projective group using the method of moving frames. Specifically, the authors derive a set of second-order differential invariants for multi-channel input functions, thus supporting RGB images and intermediate CNN features. These invariants are combined with small MLPs to produce learnable equivariant layers, which can be used as drop-in replacements for standard convolutional layers.

**Questions:**

* Can the authors report accuracy on the unwarped test?
* Can the authors add a baseline where ResNet-18 is trained with random projective data augmentations?
* can the authors discuss runtime, memory and training?
* what would happen if the network is applied to a near projection mapping? say, in a multiview setting?

**Ethical Concerns:**

["NO or VERY MINOR ethics concerns only"]

**Final Justification:**

I appreciate the authors detailed response. I was mainly impressed by their effort in swiftly implementing the point correspondence experiment which as I argued and the authors acknowledged is the more common use case for this approach. The results look promising and will likely set an important baseline for future research. I'm happy to maintain my acceptance score for this work.

**Limitations:**

the paper should put more emphasis on scalability and computational overhead

**Quality:**

3

**Strengths And Weaknesses:**

**Strengths**
(+) It's really refreshing to see a apper tacking equivariance of projective transformations. I think it's a great research topic with potentially valuable outcomes in practice
(+) Thue usage of second-order differential invariants derived from the moving frame method is elegant and novel. Removing the need to sample/discretize the group
(+) As far as i could tell, the paper is mathematically sound
(+) improvement over non equivariant baseline is significant

**Weaknesses**
(-) the evaluation design is lacking. Usually, it is common to show the vanilla (non equivariant network) performance with augmentatinos (see works like vector neurons, or G-CNNs). Also, reporting the network results on a non transformed data would give a clue regarding the results themselves -- is 51% on STL-10 considered strong?
(-) i'm also missing some analysis --  what are failure cases? do they make sense?
(-) the baseline homConv is only showed with OOM -- is there a setup where its performance could be evaluated and conpared?
(-) i'm missing a discussion of computational cost, stability is also of interest given the usage of second order derivatives
(-) classificaiton is generally not a problem where homographies are of interest. It is more suitable for point correspondences -- if the authors would have showen this it would really strenghten the paper in my view

---

> ### Author Rebuttal · Authors · 2025-07-31
>
> ### Weaknesses
>
> > **W1:** the evaluation design is lacking. Usually, it is common to show the vanilla (non equivariant network) performance with augmentatinos (see works like vector neurons, or G-CNNs). Also, reporting the network results on a non transformed data would give a clue regarding the results themselves -- is 51\% on STL-10 considered strong?
>
> **R1:**
> Thank you for the suggestion.
> We agree that comparisons with data augmentation baselines and evaluations on non-transformed data are common in other equivariant network studies.
> However, our work adopts the out-of-distribution (OOD) evaluation setup used by homConv [1], which is the most relevant prior work targeting projective equivariance and serves as our primary point of comparison.
> As prompted by your suggestion, we have added a direct numerical comparison with homConv in Tables 1 and 2 (please refer to R3 for details).
>
> This evaluation setup is designed to test whether a model can generalize to projectively transformed inputs without access to the specific distribution of test-time transformations.
> While our model incorporates prior knowledge of the symmetry group (i.e., the projective group), it does not rely on assumptions about the specific range or distribution of transformation parameters.
> In contrast, data augmentation approaches require explicitly specifying and sampling from these parameter spaces, which are often unknown or difficult to specify in practice.
>
> Our focus is on assessing the model’s ability to generalize under complex geometric transformations via built-in symmetry.
> Results on non-transformed data, where projective distortions are rare, are therefore not the primary concern, as the goal is not to optimize in-distribution performance but to test generalization under transformation shifts.
> We show that PDINet outperforms a ResNet with the same architecture in this setting, and scales more effectively than homConv, highlighting its advantages in both robustness and scalability.
>
> We believe this evaluation protocol remains consistent with prior work [1] and is well-suited for assessing the intended capabilities of projective equivariant networks.
>
> > **W2:** i'm also missing some analysis -- what are failure cases? do they make sense?
>
> **R2:**
> In our current experiments, we think the main sources of failure cases can be attributed to the following factors.
>
> Transformation-Induced Artifacts:
> Synthetic projective transformations involve interpolation and padding, which can introduce aliasing, distortions, and unnatural edges, especially under strong shearing, scaling or non-orthogonal rotations.
> Under certain transformations, these effects are especially pronounced and can degrade performance by interfering with the model’s ability to maintain equivariance.
>
> Pixel Discretization Errors:
> Our model leverages differential invariants, depending on discrete approximations of derivatives.
> As a result, image resolution may have some impact on performance.
> PDINet performs better on Proj-Imagenette than on Proj-STL-10, may partially attributed to the higher resolution enabling more accurate derivative estimation.
>
> We expect that applying PDINet to real-world data at higher resolutions with naturally occurring projective distortions would help mitigate these artifacts and better demonstrate its full potential.
>
> > **W3:** the baseline homConv is only showed with OOM -- is there a setup where its performance could be evaluated and conpared?
>
> **R3:**
> To enable a direct numerical comparison with homConv, we follow the original network structure used in [1] and reduce the number of samples to avoid out-of-memory (OOM) issues.
> The experimental results are shown in Tables 1 and 2.
> Due to its restriction on network depth, homConv struggles to perform well on image datasets with higher resolutions.
>
> **Table 1:** Test accuracy (\%) on Proj-STL-10 after training on STL-10.
> | Model                      | Accuracy  | # Params  |
> |----------------------------|----------------|------------------|
> | ResNet-18                  | $39.73 \pm 0.33$ | $11.18$ M  |
> | homConv                    | $21.26$          | $376$ K  |
> | PDINet (ours)              | $51.12 \pm 0.47$ | $8.90$ M  |
>
> **Table 2:** Test accuracy (\%) on Proj-Imagenette after training on Imagenette.
> | Model                      | Accuracy  | # Params  |
> |----------------------------|----------------|------------------|
> | ResNet-18                  | $65.64 \pm 0.39$ | $11.18$ M  |
> | homConv                    | $26.57$          | $376$ K  |
> | PDINet (ours)              | $71.30 \pm 0.45$ | $8.90$ M  |
>
> > **W4:** i'm missing a discussion of computational cost, stability is also of interest given the usage of second order derivatives
>
> **R4:**
> Our projective equivariant layer scales linearly in both time and space complexity with respect to the input size.
> Specifically, its computation include estimating derivatives, computing and combining differential invariants, each with linear complexity.
> We use torchstat to numerically measure the memory usage and the FLOPs of a single layer required for model inference.
> As shown in Table 3, both the memory usage and the FLOPs scale linearly with the input size.
>
> PDINet consistently shows stable training behavior.
> We plan to include convergence plots in future revisions.
>
> **Table 3**: The FLOPs and memory occupation of a projective equivariant layer.
> | Input Size     | Memory (MB) | FLOPs          |
> |----------------|-------------|----------------|
> |$32 \times 32$  | $0.02$      |$5.53\times10^4$|
> |$64 \times 64$  | $0.09$      |$2.21\times10^5$|
> |$128\times128$  | $0.38$      |$8.84\times10^5$|
> |$256\times256$  | $1.50$      |$3.54\times10^6$|
> |$512\times512$  | $6.00$      |$1.42\times10^7$|
>
> > **W5:** classificaiton is generally not a problem where homographies are of interest. It is more suitable for point correspondences -- if the authors would have showen this it would really strenghten the paper in my view
>
> **R5:**
> Thank you for the insightful comment.
> We agree that classification may not be the task most inherently associated with projective transformations.
> However, it remains a common and practical benchmark for evaluating equivariant architectures, as also adopted by homConv [1].
> Since object categories should remain unchanged under projective distortions, classification still serves as a meaningful testbed for assessing a model's ability to generalize under such transformations.
>
> That said, we fully agree that tasks such as point correspondences may better reflect the utility of projective equivariance in real-world geometric transformation.
> Motivated by your suggestion, we conduct a preliminary experiment to evaluate the effectiveness of our proposed PDINet in the context of keypoint matching.
> Specifically, we integrate PDINet as the backbone into the REKD [2] framework for keypoint detection.
> To reduce confounding factors, we simplify the pipeline by removing the orientation estimation branch and retain only the keypoint detection component.
> We use a three-layer PDINet and compare it against a CNN baseline with the same architecture.
> The number of channels is adjusted to keep the parameter counts on the same order of magnitude, with PDINet using fewer parameters overall.
>
> We evaluate both models on the viewpoint split of the HPatches dataset, which includes 59 scenes, each with a reference image and five target images captured from different viewpoints, inducing projective distortions.
> Both models are trained for 20 epochs using the AdamW optimizer with a cosine learning rate schedule (initial learning rate 1e-2).
> Following [2], we adopt the same data construction and loss, and apply a downsampling pyramid with scaling factor $1.2$ during training.
> At inference, we use a symmetric downsampling/upsampling pyramid with scale factor $\sqrt{2}$ and an identity branch.
> The number of down-sampling and up-sampling are both set to 2.
>
> We report the Repeatability metric, which measures the consistency of keypoint detection under viewpoint changes.
> A higher value indicates better robustness.
> As shown in Table 4, PDINet achieves higher repeatability than the baseline while using fewer parameters, suggesting that built-in projective equivariance enhances geometric consistency in keypoint detection.
>
> We appreciate your suggestion, which has helped us demonstrate the broader applicability of PDINet beyond classification.
> We believe this direction is promising and worth further exploration in future work.
>
> **Table 4**: Keypoint detection results on the HPatches viewpoint split.
> | Model    | Repeatability ↑ | # Params |
> | -------- | --------------- | ------------ |
> | CNN      | 42.1            | 8.3 K         |
> | PDINet   | 45.2            | 3.2 K         |
>
> ### Questions
>
> Responses to Q1 and Q2 can be found in our reply to W1.
>
> Responses to Q3 and limitations can be found in our reply to W4.
>
> > **Q4:** what would happen if the network is applied to a near projection mapping? say, in a multiview setting?
>
> **A4:**
> Thank you for the question.
> In multiview settings involving planar objects, transformations between two viewpoints are related by projective mappings, making PDINet naturally suitable for such scenarios.
> For instance, the HPatches dataset used in our keypoint matching experiment (Table 4) contains planar scenes captured from different angles, and PDINet performs well under these projective variations.
>
> While exact equivariance may not strictly hold in cases involving non-planar objects, we believe that the projective inductive bias still contributes to improved robustness by approximately preserving geometric structure under near projective transformations.
> Exploring such extensions, including more general 3D settings, is a promising direction for future work.
>
> [1] MacDonald et al. Enabling equivariance for arbitrary Lie groups (2022)
>
> [2] Lee et al. Self-Supervised Equivariant Learning for Oriented Keypoint Detection (2022)

---

> ### Comment · Area_Chair_Jh9D · 2025-08-07
>
> Dear reviewer gqa7, thanks for your time reviewing this paper. Although your opinion might be clear, according to this year's NeurIPS policy, it is required to have reviewer-author discussions before submitting the mandatory acknowledgement. Could you please take some time to comment on the authors' responses? Thanks!

---

### Note · Authors · 2025-08-13

We sincerely thank the reviewers for their thoughtful feedback and constructive suggestions.

**Summary of contributions and strengths recognized by reviewers**

This work achieves projective equivariance in a scalable way, addressing an important and challenging problem (Reviewers gqa7, 6v2q, VbGP, and rwYg) and making a significant contribution to the field of equivariant deep learning (Reviewer 6v2q).
It is built on a mathematically sound framework of second-order differential invariants derived via the moving frame method (Reviewers gqa7, 6v2q, and VbGP), removing the need to sample or discretize the group (Reviewer gqa7).
The resulting model is theoretically grounded and empirically scalable (Reviewers gqa7, 6v2q, and VbGP), showing substantial improvements over the non-equivariant baseline (Reviewer gqa7).
Reviewers also praised the paper’s clear structure (Reviewer 6v2q), thorough explanation of the mathematical background (Reviewers 6v2q and rwYg), and originality (Reviewers VbGP and rwYg).

**Rebuttal-stage updates and additions**

During the rebuttal and discussion period, we provided additional experiments and clarifications following reviewers’ suggestions to strengthen the work:

- Direct comparison with the main counterpart homConv, enabling a clearer evaluation (Reviewers gqa7, 6v2q, and VbGP).
- Inclusion of ResNet-18 with projective data augmentation as an additional baseline, providing a more comprehensive assessment (Reviewers gqa7, 6v2q, VbGP, and rwYg).
- Application to keypoint detection in the context of keypoint matching, demonstrating practical benefits of built-in projective equivariance (Reviewers gqa7, VbGP, and rwYg).
- Measurement and analysis of computational complexity (Reviewers gqa7, 6v2q, and VbGP).
- Measurement and analysis of equivariance error (Reviewers 6v2q and VbGP).
- Clarification of derivative estimation and the impact of discretization error (Reviewer 6v2q), analysis of failure cases (Reviewer gqa7), and explanation of the geometric meaning of projective transformations (Reviewer VbGP).
- Discussion of extensions to Möbius transformations (Reviewer rwYg) and to multi-view settings (Reviewer gqa7).

We will incorporate these updates and clarifications into the revised version.
We are grateful for the constructive engagement throughout the review process, which has helped us significantly enhance the quality and clarity of the paper.

---

### Decision · Program_Chairs · 2025-09-17

**Decision:**

Accept (spotlight)

**Comment:**

[Summary]

This paper proposes a network architecture with projective equivariance (equivariant to homographies). The main contributions of this paper are two-fold: (i) theoretical contribution: a set of second-order fundamental differential invariants for the projective group; (ii) practical network design: sampling-free (enabling scalability), easy drop-in replacement for standard convolution layers, and showing good empirical results.


[Strengths]

- The paper tackles a problem that is very underexplored, and proposes an idea with high originality.
- Theoretically, the paper has solid theoretical derivations and proposes a novel network design based on that.
- Empirically, the proposed framework has good scalability and robustness, with good experimental results.
- The paper is well-written.


[Weaknesses]

The reviewers speak very highly of the work. Most of the questions and concerns are well-addressed by the authors during the rebuttal period. If I have to point out some weaknesses, it could only be that: at the current stage, the problem and framework are not yet fully developed to make real-world impacts. But I think all research problems need to go through this beginning stage.

Also, as a lot of additional experiments and clarifications are added in the discussion period, I hope the authors can add them to the paper in their final revisions.


[Reasons for decision]

This paper received very high comments from the reviewers: proposing almost the first framework that can truly tackle a difficult problem, with both theoretical and practical contributions. The reviewers would also like to strongly champion this work.

As I also work in this field, I understand that having a work like this is non-trivial: many theoretically sound works struggle in experiments, while many works with good performances lack theoretical justifications. The small concerns/questions are well-addressed by the authors during the rebuttal period, with promising experimental results and detailed analysis.

Therefore, I recommend acceptance, and I think it deserves at least a spotlight. Either spotlight or oral makes sense to me. (I'm a bit unsure about the criteria between spotlight and oral.) I personally would like to recommend oral, as I prefer this type of risk-taking works, even if its real-world applications are not fully developed.


[Discussion summary]

Initially, the paper received an average score of borderline acceptance. All reviewers appreciate the originality of the work. Certain concerns were around: (i) lack of clarity on some technical details; (ii) the experimental comparisons/analysis were relatively weak; (iii) the task chosen in the experiments (classification) was not fully addressing the strength of the proposed method; (iv) not demonstrating enough capability/potential of real-world applications.

During the discussion period, the authors have added a number of experiments and clarifications to address the reviewers' comments. The reviewers have raised their scores to all acceptance, with multiple reviews strongly championing this work. I also appreciate all their efforts made in the discussions.


[SAC Update for final oral-spotlight calibration]

This submission explores how to build projective invariants, with applications to computer vision. Its main technical contribution is a novel architecture based on so-called differential invariants, which leverage relationships between group actions across several derivatives of the input field.
Overall, this is a strong submission that tackles an under-explored problem, and where consensus quickly emerged amongst reviewers and AC. While highly promising, the method currently applies only to specific projective settings, somewhat limiting its significance. All in all, we recommend a spotlight presentation.